# Self-Adjusting Variable Neighborhood Search Algorithm for Near-Optimal k-Means Clustering

**Lev Kazakovtsev \*, Ivan Rozhnov, Aleksey Popov and Elena Tovbis**

Reshetnev Siberian State University of Science and Technology, Institute of Informatics and Telecommunications, Krasnoyarskiy Rabochiy av. 31, 660037 Krasnoyarsk, Russia; ris2005@mail.ru (I.R.); vm_popov@sibsau.ru (A.P.); sibstu2006@rambler.ru (E.T.)

\* Correspondence: levklevk@gmail.com

**Abstract:** The k-means problem is one of the most popular models in cluster analysis that minimizes the sum of the squared distances from clustered objects to the sought cluster centers (centroids). The simplicity of its algorithmic implementation encourages researchers to apply it in a variety of engineering and scientific branches. Nevertheless, the problem is proven to be NP-hard which makes exact algorithms inapplicable for large scale problems, and the simplest and most popular algorithms result in very poor values of the squared distances sum. If a problem must be solved within a limited time with the maximum accuracy, which would be difficult to improve using known methods without increasing computational costs, the variable neighborhood search (VNS) algorithms, which search in randomized neighborhoods formed by the application of greedy agglomerative procedures, are competitive. In this article, we investigate the influence of the most important parameter of such neighborhoods on the computational efficiency and propose a new VNS-based algorithm (solver), implemented on the graphics processing unit (GPU), which adjusts this parameter. Benchmarking on data sets composed of up to millions of objects demonstrates the advantage of the new algorithm in comparison with known local search algorithms, within a fixed time, allowing for online computation.

**Keywords:** cluster analysis; k-means; variable neighborhood search; agglomerative clustering; GPU

## 1. Introduction

### 1.1. Problem Statement

The aim of a clustering problem solving is to divide a given set (sample) of objects (data vectors) into disjoint subsets, called clusters, so that each cluster consists of similar objects, and the objects of different clusters have significant dissimilarities [1,2]. The clustering problem belongs to a wide class of unsupervised machine learning problems. Clustering models involve various similarity or dissimilarity measures. The k-means model with the squared Euclidean distance as a dissimilarity measure is based exclusively on the maximum similarity (minimum sum of squared distances) among objects within clusters.

Clustering methods can be divided into two main categories: hierarchical and partitioning [1,3]. Partitioning clustering, such as k-means, aims at optimizing the clustering result in accordance with a pre-defined objective function [3].

The k-means problem [4,5], also known as minimum sum-of-squares clustering (MSSC), assumes that the objects being clustered are described by numerical features. Each object is represented by a point in the feature space $\mathbb{R}^d$ (data vector). It is required to find a given number $k$ of cluster centers

(called centroids), such as to minimize the sum of the squared distances from the data vectors to the nearest centroid.

Let $A_1, \dots, A_N \in \mathbb{R}^d$ be data vectors, $N$ be the number of them, and $S = \{X_1, \dots, X_k\} \subset \mathbb{R}^d$ be the set of sought centroids. The objective function (sum of squared errors, SSE) of the k-means optimization problem formulated by MacQueen [5] is:

$$SSE(X_1, \dots, X_k) = SSE(S) = \sum_{i=1}^{N} min_{X \in \{X_1, \dots, X_k\}} \|A_i - X\|^2 \rightarrow min_{X_1, \dots, X_k \in \mathbb{R}^d}. \tag{1}$$

Here, $\|\cdot\|$ is the Euclidean distance, integer $k$ must be known in advance.

A cluster in the k-means problem is a subset of data vectors for which the specified centroid is the nearest one:

$$C_j = \left\{ A_i, i = \overline{1,N} \middle| \|A_i - X_j\| = min_{X \in \{X_1, \dots, X_k\}} \|A_i - X\| \right\}, \quad j = \overline{1,k}.$$

We assume that a data vector cannot belong to two clusters at the same time. At an equal distance for several centroids, the question of assignment to a cluster can be solved by clustering algorithms in different ways. For example, a data vector belongs to a cluster lower in number:

$$
\begin{aligned}
C_j = \Big\{ A_i, i = \overline{1,N} \Big| \nexists j' = \overline{1,k} : \|A_i - X_{j'}\| \\
< \|A_i - X_j\| \text{ or } \left( \|A_i - X_{j'}\| = \|A_i - X_j\| \text{ and } j' > j \right) \Big\}, \\
j = \overline{1,k}.
\end{aligned}
\tag{2}
$$

Usually, for practical problems with sufficiently accurate measured values of data vectors, the assignment to a specific cluster is not very important.

The objective function may also be formulated as follows:

$$SSE(X_1, \dots, X_k) = \sum_{j=1}^{k} \sum_{i=\overline{1,N}:A_i \in C_j} \|A_i - X_j\|^2 \rightarrow min_{X_1, \dots, X_k \in \mathbb{R}^d}, \cdot \tag{3}$$

or

$$SSE(C_1, \dots, C_k) = \sum_{j=1}^{k} \sum_{i=\overline{1,N}:A_i \in C_j} \|A_i - X_j\|^2 \rightarrow min_{C_1, \dots, C_k \subset \{A_1, \dots, A_N\}}. \tag{4}$$

Equations (3) and (4) correspond to continuous and discrete statements of our problem, respectively.

Such clustering problem statements have a number of drawbacks. In particular, the number of clusters $k$ must be given in advance, which is hardly possible for the majority of practically important problems. Furthermore, the adequacy of the result in the case of a complex cluster shapes is questionable (this model is proved to work fine with the ball-shaped clusters [6]). The result is sensitive to the outliers (standalone objects) [7,8] and depends on the chosen distance measure and the data normalization method. This model does not take into account the dissimilarity between the objects in different clusters, and the application of the k-means model results in some solution $X_1, \dots,$ $X_k$ even in the cases with no cluster structure in the data [9,10]. Moreover, the NP-hardness [11,12] of the problem makes the exact methods [6] applicable only for very small problems.

Nevertheless, the simplicity of the most commonly used algorithmic realization as well as the interpretability of the results make the k-means problem the most popular clustering model. Developers' efforts are focused on the design of heuristic algorithms that provide acceptable and attainable values of the objective function.

*1.2. State of the Art*

The most commonly used algorithm for solving problem (1) is the Lloyd's procedure proposed in 1957 and published in 1982 [4], also known as the k-means algorithm, or alternate location-allocation (ALA) algorithm [13,14]. This algorithm consists of two simple alternating steps, the first of which solves the simplest continuous (quadratic) optimization problem (3), finding the optimal positions of the centroids $X_1, \ldots, X_k$ for a fixed composition of clusters. The second step solves the simplest combinatorial optimization problem (4) by redistributing data vectors between clusters at fixed positions of the centroids. Both steps aim at minimizing the SSE. Despite the theoretical estimation of the computational complexity being quite high [15–17], in practice, the algorithm quickly converges to a local minimum. The algorithm starts with some initial solution $S = \{X_1, \ldots, X_k\}$, for instance, chosen at random, and its result is highly dependent on this choice. In the case of large-scale problems, this simple algorithm is incapable of obtaining the most accurate solutions.

Various clustering models are widely used in many engineering applications [18,19], such as energy loss detection [20], image segmentation [21], production planning [22], classification of products such as semiconductor devices [23], recognition of turbulent flow patterns [24], and cyclical disturbance detection in supply networks [25]. Clustering is also used as a preprocessing step for the supervised classification [26].

In [27], Naranjo et al. use various clustering approaches including the k-means model for automatic classification of traffic incidents. The approach proposed in [28] uses the k-means clustering model for the optimal scheduling of public transportation. Sesham et al. [29] use factor analysis methods in a combination with the k-means clustering for detecting cluster structures in transportation data obtained from the interview survey. Such data include the geographic information (home addresses) and general route information. The use of GPS sensors [30] for collecting traffic data provides us with large data arrays for such problems as the travel time prediction, traffic condition recognition [31], etc.

The k-means problem can be classified as a continuous location problem [32,33]: it is aimed at finding the optimal location of centroids in a continuous space.

If we replace squared distances with distances in (1), we deal with the very similar continuous k-median location problem [34] which is also a popular clustering model [35]. The k-medoids [36,37] problem is its discrete version where cluster centers must be selected among the data vectors only, which allows us to calculate the distance matrix in advance [38]. The k-median problem was also formulated as a discrete location problem [39] on a graph. The similarity of these NP-hard problems [40,41] enables us to use similar approaches to solve them. In the early attempts to solve the k-median problem (its discrete version) by exact methods, researchers used a branch and bound algorithm [42–44] for solving very small problems.

Metaheuristic approaches, such as genetic algorithms [45], are aimed at finding the global optimum. However, in large-scale instances, such approaches require very significant computational costs, especially if they are adapted to solving continuous problems [46].

With regard to discrete optimization problems, local search methods, which include Lloyd's procedure, have been developed since the 1950s [47–50]. These methods have been successfully used to solve location problems [51,52]. The progress of local search methods is associated with both new algorithmic schemes and new theoretical results in the field of local search [50].

A standard local search algorithm starts with some initial solution $S$ and goes to a neighboring solution if this solution turns out to be superior. Moreover, finding the set of neighbor solutions $n(S)$ is the key issue. Elements of this set are formed by applying a certain procedure to a solution $S$. At each local search step, the neighborhood function $n(S)$ specifies the set of possible search directions. Neighborhood functions can be very diverse, and the neighborhood relation is not always symmetric [53,54].

For a review of heuristic solution techniques applied to k-means and k-median problems, the reader can refer to [32,55,56]. Brimberg, Drezner, and Mladenovic and Salhi [57–59] presented local search approaches including the variable neighborhood search (VNS) and concentric search. In [58], Drezner et al. proposed heuristic procedures including the genetic algorithm (GA), for rather small

data sets. Algorithms for finding the initial solution for the Lloyd's procedure [60,61] are aimed at improving the average resulting solution. For example, in [62], Bhusare et al. propose an approach to spread the initial centroids uniformly so that the distance among them is as far as possible. The most popular kmeans++ initialization method introduced by Arthur and Vassilvitskii [60] is a probabilistic implementation of the same idea. An approach proposed by Yang and Wang [63] improves the traditional k-means clustering algorithm by choosing initial centroids with a min-max similarity. Gu et al. [7] provide a density-based initial cluster center selection method to solve the problem of outliers. Such smart initialization algorithms reduce the search area for local search algorithms in multi-start modes. Nevertheless, they do not guarantee an optimal or near optimal solution of the problem (1).

Many authors propose approaches based on reducing the amount of data [64]: simplification of the problem by random (or deterministic) selection of a subset of the initial data set for a preliminary solution of the k-means problem, and using these results as an initial solution to the k-means algorithm on the complete data set [65–67]. Such aggregation approaches, summarized in [68], as well as reducing the number of the data vectors [69], enable us to solve large-scale problems within a reasonable time. However, such approaches lead to a further reduction in accuracy. Moreover, many authors [70,71] name their algorithm "exact" which does not mean the ability to achieve an exact solution of (1). In such algorithms, the word "exact" means the exact adherence to the scheme of the Lloyd's procedure, without any aggregation, sampling, and relaxation approaches. Thus, such algorithms may be faster than the Lloyd's procedure due to the use of triangle inequality, storing the results of distance calculations in multidimensional data sets or other tricks [72], however they are not intended to get the best value of (1). In our research, aimed at obtaining the most precise solutions, we consider only the methods which estimate the objective function (1) directly, without aggregation or approximation approaches.

The main idea of the variable neighborhood search algorithms proposed by Hansen and Mladenovic [73–75] is the alternation of neighborhood functions $n(S)$. Such algorithms include Lloyd's procedure, which alternates finding a locally optimal solution of a continuous optimization problem (3) with a solution of a combinatorial problem (4). However, as applied to the k-means problem, the VNS class traditionally involves more complex algorithms.

The VNS algorithms are used for a wide variety of problems [3,76,77] including clustering [78] and work well for solving k-means and similar problems [50,79–82].

Agglomerative and dissociative procedures are separate classes of clustering algorithms. Dissociative (divisive) procedures [83] are based on splitting clusters into smaller clusters. Such algorithms are commonly used for small problems due to their high computing complexity [83–85], most often in hierarchical clustering models. The agglomerative approach is the most popular in hierarchical clustering, however, it is also applied in other models of cluster analysis. Agglomerative procedures [86–90] combine clusters sequentially, i.e., in relation to the k-means problem, they sequentially remove centroids. The elements of the clusters, related to the removed centroids, are redistributed among the remaining clusters. The greedy strategies are used to decide which clusters are most similar to be merged together [3] at each iteration of the agglomerative procedure. An agglomerative procedure starts with some solution $S$ containing an excessive number of centroids and clusters $k + r$, where integer $r$ is known in advance or chosen randomly. The $r$ value (number of excessive centroids in the temporary solution) is the most important parameter of the agglomerative procedure. Some algorithms, including those based on the k-means model [91], involve both the agglomerative and dissociative approaches. Moreover, such algorithms are not aimed at achieving the best value of the objective function (1), and their accuracy is not high in this sense.

*1.3. Research Gap*

Many transportation and other problems (e.g., clustering problems related to computer vision) require online computation within a fixed time. As mentioned above, Lloyd's procedure, the most popular k-means clustering algorithm, is rather fast. Nevertheless, for specific data sets including

geographic/geometrical data, this algorithm results in a solution which is very far from the global minimum of the objective function (1), and the multi-start operation mode does not improve the result significantly. More accurate k-means clustering methods are much slower. Nevertheless, recent advances in high-performance computing and the use of massively parallel systems enable us to work through a large amount of computation using the Lloyd's procedure embedded into more complex algorithmic schemes. Thus, the demand for clustering algorithms that compromise on the time spent for computations and the resulting objective function (1) value is apparent. Nevertheless, in some cases, when solving problem (1), it is required to obtain a result (a value of the objective function) within a limited fixed time, which would be difficult to improve on by known methods without a significant increase in computational costs. Such results are required if the cost of error is high, as well as for evaluating faster algorithms, as reference solutions.

Agglomerative procedures, despite their relatively high computational complexity, can be successfully integrated into more complex search schemes. They can be used as a part of the crossover operator of genetic algorithms [46,88] and as a part of the VNS algorithms. Moreover, such algorithms are a compromise between the solution accuracy and time costs. In this article, by accuracy, we mean exclusively the ability of the algorithm (solver) to obtain the minimum values of the objective function (1).

The use of VNS algorithms, that search in the neighborhoods, formed by applying greedy agglomerative procedures to a known (current) solution $S$, enables us to obtain good results in a fixed time acceptable for interactive modes of operation. The selection of such procedures, their sequence and their parameters remained an open question. The efficiency of such procedures has been experimentally shown on some test and practical problems. Various versions of VNS algorithms based on greedy agglomerative procedures differ significantly in their results which makes such algorithm difficult to use in practical problems. It is practically impossible to forecast the relative performance of a specific VNS algorithm based on such generalized numerical features of the problem as the sample size and the number of clusters. Moreover, the efficiency of such procedures depends on their parameters. However, the type and nature of this dependence has not been studied.

### 1.4. Our Contribution

In this article, we systematize approaches to the construction of search algorithms in neighborhoods, formed by the use of greedy agglomerative procedures.

In this work, we proceeded from the following assumptions:

(a) The choice of parameter $r$ value (the number of excessive centroids, see above) of the greedy agglomerative heuristic procedure significantly affects the efficiency of the procedure.

(b) Since it is hardly possible to determine the optimal value of this parameter based on such numerical parameters of the k-means problem as the number of data vectors and the number of clusters, reconnaissance (exploratory) search with various values of $r$ can be useful.

(c) Unlike the well-known VNS algorithms that use greedy agglomerative heuristic procedures with an increasing value of the parameter $r$, a gradual decrease in the value of this parameter may be more effective.

Based on these assumptions, we propose a new VNS algorithm involving greedy agglomerative procedures for the k-means problem, which, by adjusting the initial $r$ parameter of such procedures, enables us to obtain better results in a fixed time which exceed the results of known VNS algorithms. Due to self-adjusting capabilities, such an algorithm should be more versatile, which should increase its applicability to a wider range of problems in comparison with known VNS algorithms based on greedy agglomerative procedures.

### 1.5. Structure of this Article

The rest of this article is organized as follows. In Section 2, we present an overview of the most common local search algorithms for k-means and similar problems, and introduce the notion

of neighborhoods SWAP$_r$ and GREEDY$_r$. It is shown experimentally that the search result in these neighborhoods strongly depends on the neighborhood parameter $r$ (the number of simultaneously alternated or added centroids). In addition, we present a new VNS algorithm which performs the local search in alternating GREEDY$_r$ neighborhoods with the decreasing value of $r$ and its initial value estimated by a special auxiliary procedure. In Section 3, we describe our computational experiments with the new and known algorithms. In Section 4, we consider the applicability of the results on the adjustment of the GREEDY$_r$ neighborhood parameter in algorithmic schemes other than VNS, in particular, in evolutionary algorithms with a greedy agglomerative crossover operator. The conclusions are given in Section 5.

## 2. Materials and Methods

For constructing a more efficient algorithm (solver), we used a combination of such algorithms as Lloyd's procedure, greedy agglomerative clustering procedures, and the variable neighborhood search. The most computationally expensive part of this new algorithmic construction, Lloyd's procedure, was implemented on graphic processing units (GPU).

*2.1. The Simplest Approach*

Lloyd's procedure, the simplest and most popular algorithm for solving the k-means problem, is described as follows (see Algorithm 1).

---

**Algorithm 1.** *Lloyd*($S$)

---

**Require:** Set of initial centroids $S = \{X_1, \ldots, X_k\}$. If $S$ is not given, then the initial centroids are selected randomly from the set of data vectors $\{A_1, \ldots, A_N\}$.

**repeat**

1. For each centroid $X_j$, $j = \overline{1, k}$, define its cluster in accordance with (2); // *I.e. assign each data vector to the nearest centroid*

2. For each cluster $C_j$, $j = \overline{1, k}$, calculate its centroid as follows:

$$X_j = \frac{\sum_{i \in (\overline{1,N}):A_i \in C_j} A_i}{|C_j|}.$$

**until** all centroids stay unchanged.

---

Formally, the k-means problem in its formulation (1) or (3) is a continuous optimization problem. With a fixed composition of clusters $C_j$, the optimal solution is found in an elementary way, see Step 2 in Algorithm 1, and this solution is the local optimum of the problem in terms of the continuous optimization theory, i.e., local optimum in the $\varepsilon$-neighborhood. A large number of such optima forces the algorithm designers to systematize their search in some way. The first step of Lloyd's algorithm solves a simple combinatorial optimization problem (3) on the redistribution of data vectors among clusters, that is, it searches in the other neighborhood.

The simplicity of Lloyd's procedure enables us to apply it to a wide range of problems, including face detection, image segmentation, signal processing and many others [92]. Frackiewicz et al. [93] presented a color quantization method based on downsampling of the original image and k-means clustering on a downsampled image. The k-means clustering algorithm used in [94] was proposed for identifying electrical equipment of a smart building. In many cases, researchers do not distinguish between the k-means model and the k-means algorithm, as Lloyd's procedure is also called. Nevertheless, the result of Lloyd's procedure may differ from the results of other more advanced algorithms many times in the objective function value (1). For finding a more accurate solution, a wide range of heuristic methods were proposed [55]: evolutionary and other bio-inspired algorithms, as well as local search in various neighborhoods.

Modern scientific literature offers many algorithms to speed up the solution of the k-means problem. Algorithm named k-indicators [95] promoted by Chen et al. is a semi-convex-relaxation algorithm for approximate solution of big-data clustering problems. In the distributed implementation of the k-means algorithm proposed in [96], the algorithm considers a set of agents, each of which is equipped with a possibly high-dimensional piece of information or set of measurements. In [97,98], the researchers improved algorithms for the data streams. In [99], Hedar et al. present a hierarchical k-means method for better clustering performance in the case of big data problems. This approach enables us to mitigate the poor scaling behavior with regard to computing time and memory requirements. Fast adaptive k-means subspace clustering algorithm with an adaptive loss function for high-dimensional data was proposed by Wang et al. [100]. Nevertheless, the usage of the massively parallel systems is the most efficient way to achieve the most significant acceleration of computations, and the original Lloyd's procedure (Algorithm 1) can be seamlessly parallelized on such systems [101,102].

Metaheuristic approaches for the k-means and similar problems include genetic algorithms [46,103,104], the ant colony clustering hybrid algorithm proposed in [105], particle swarm optimization algorithms [106]. Almost all of these algorithms in one way or another use the Lloyd's procedure or other local search procedures. Our new algorithm (solver) is not an exception.

### 2.2. Local Search in SWAP Neighborhoods

Local search algorithms differ in forms of neighborhood function $n(S)$. A local minimum in one neighborhood may not be a local minimum in another neighborhood [50]. The choice of a neighborhood of lower cardinality leads to a decrease in the complexity of the search step, however, a wider neighborhood can lead to a better local minimum. We have to find a balance between these conflicting requirements [50].

A popular idea when solving k-means, k-medoids, k-median problems is to search for a better solution in SWAP neighborhoods. This idea was realized, for instance, in the J-means procedure [80] proposed by Hansen and Mladenovic, and similar I-means algorithm [107]. In SWAP neighborhoods, the set $n(S)$ is the set of solutions obtained from $S$ by replacing one or more centroids with some data vectors.

Let us denote the neighborhood, where $r$ centroids must be simultaneously replaced, by $SWAP_r(S)$. The $SWAP_r$ neighborhood search can be regular (all possible substitutions are sequentially enumerated), as in the J-means algorithm, or randomized (centroids and data vectors for replacement are selected randomly). In both cases, the search in the SWAP neighborhood always alternates with the Lloyd's procedure: if an improved solution is found in the SWAP neighborhood, the Lloyd's procedure is applied to this new solution, and then the algorithm returns to the SWAP neighborhood search. Except for very small problems, regular search in SWAP neighborhoods, with the exception of the $SWAP_1$ neighborhood and sometimes $SWAP_2$, is almost never used due to the computational complexity: in each of iterations, all possible replacement options must be tested. A randomized search in $SWAP_r$ neighborhoods can be highly efficient for sufficiently large problems, which can be demonstrated by the experiment described below. Herewith, the correct choice of $r$ is of great importance.

As can be seen on Figure 1, for various problems from the clustering benchmark repository [108,109], the best results are achieved with different values of $r$, although in general, such a search provides better results in comparison with Lloyd's procedure. Our computational experiments are described in detail in Sections 2.5–2.7.

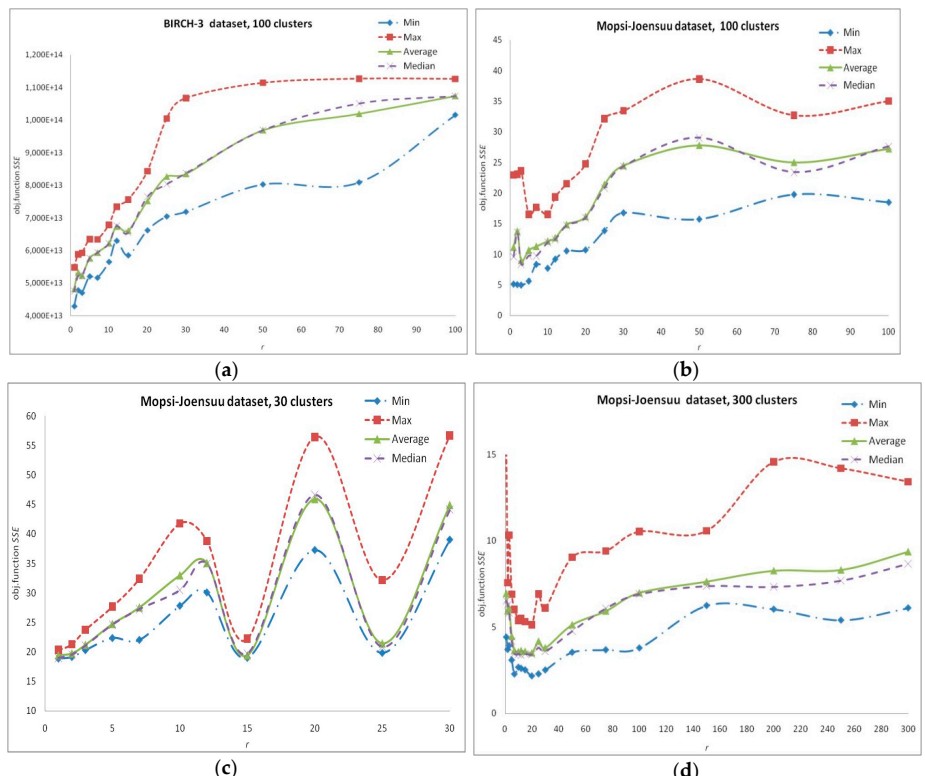

**Figure 1.** Search in SWAP*r* neighborhoods. Dependence of the result on *r*: (**a**) BIRCH3 data set, 100 clusters, $10^5$ data vectors, time limitation 10 s; (**b–d**) Mopsi-Joensuu data set, 30, 100 and 300 clusters, 6014 data vectors, time limitation 5 s.

## 2.3. Agglomerative Approach and GREEDY$_r$ Neyborhoods

When solving the k-means and similar problems, the agglomerative approach is often successful. In [86], Sun et al. propose a parallel clustering method based on MapReduce model which implements the information bottleneck clustering (IBC) idea. In the IBC and other agglomerative clustering algorithms, clusters are sequentially removed one-by-one, and objects are redistributed among the remaining clusters. Alp et al. [88] presented a genetic algorithm for facility location problems, where evolution is facilitated by a greedy agglomerative heuristic procedure. A genetic algorithm with a faster greedy heuristic procedure for clustering and location problems was also proposed in [90]. In [46], two genetic algorithm approaches with different crossover procedures are used to solve k-median problem in continuous space.

Greedy agglomerative procedures can be used as independent algorithms, as well as being embedded into genetic operators [110] or VNS algorithms [79]. The basic greedy agglomerative procedure for the k-means problem can be described as follows (see Algorithm 2).

---

**Algorithm 2.** *BasicGreedy(S)*

---

**Require:** Set of initial centroids $S = \{X_1, \ldots, X_K\}$, $K > k$, required final number of centroids $k$.
$S \leftarrow Lloyd(S)$;
**while** $|S| > k$ **do**
　　　　**for** $i = \overline{1, K}$ **do**
　　　　　　　$F_i \leftarrow SSE(S \setminus \{X_i\})$;
　　　　**end for**
　　　　Select a subset $S' \subset S$ of $r_{toremove}$ centroids with the minimum values of the corresponding
　　　　variables $F_i$; // *By default,* $r_{toremove} = 1$.
　　　　　$S \leftarrow Lloyd(S \setminus S')$;
**end while**.

---

In its most commonly used version, with $r_{toremove} = 1$, this procedure is rather slow for large-scale problems. It tries to remove the centroids one-by-one. At each iteration, it eliminates such centroids that their elimination results in the least significant increase in the SSE value. Further, this procedure involves the Lloyd's procedure which can be also slow in the case of rather large problems with many clusters. To improve the performance of such a procedure, the number of simultaneously eliminated centroids can be calculated as $r_{toremove} = \max\{1, (|S| - k) \cdot r_{coef}\}$. In [90], Kazakovtsev and Antamoshkin used the elimination coefficient value $r_{coef} = 0.2$. This means that at each iteration, up to 20% of the excessive centroids are eliminated, and such values are proved to make the algorithm faster. In this research, we use the same value.

In [79,90,110], the authors embed the *BasicGreedy*() procedure into three algorithms which differ in $r$ value only. All of these algorithms can be described as follows (see Algorithm 3):

---

**Algorithm 3.** *Greedy* (*S*,*S*₂,*r*)

---

**Require:** Two sets of centroids $S$, $S_2$, $|S| = |S_2| = k$, the number of centroids $r$ of the solution $S_2$ which are used to obtain the resulting solution, $r \in \overline{\{1, k\}}$.

**For** $i = \overline{1, n_{repeats}}$ **do**
1. Select a subset $S' \subset S_2 : \ |S'| = r$.
　　2. $S' \leftarrow BasicGreedy(S \cup S')$;
　　3. **if** $SSE(S') < SSE(S)$ **then** $S \leftarrow S'$ **end if**;
**end for**
**return** $S$.

---

Such procedures use various values of $r$ from 1 up to $k$. If $r = 1$ then the algorithm selects a subset (actually, a single element) of $S_2$ regularly: $\{X_1\}$ in the first iteration, $\{X_2\}$ in the second one, etc. In this case, $n_{repeats} = k$. If $r = k$ then obviously $S' = S_2$, and $n_{repeat} = 1$. Otherwise, $r$ is selected randomly, $r \in \overline{\{2, k-1\}}$, and $n_{repeats}$ depends on $r$: $n_{repeats} = \max\{1, [k/r]\}$.

If the solution $S_2$ is fixed, then all possible results of applying the *Greedy*(*S*,*S*₂,*r*) procedure form a neighborhood of the solution $S$, and $S_2$ as well as $r$ are parameters of such a neighborhood. If $S_2$ is a randomly chosen locally optimal solution obtained by $Lloyd(S_2')$ procedure applied to a randomly chosen subset $S_2' \subset \{A_1, \ldots, A_N\}$, $|S_2'| = k$, then we deal with a randomized neighborhood.

Let us denote such a neighborhood by GREEDY$_r$(*S*). Our experiments in Section 3 demonstrate that the obtained result of the local search in GREEDY$_r$ neighborhoods strongly depends on $r$.

## 2.4. Variable Neighborhood Search

The dependence of the local search result on the neighborhood selection reduces if we use a certain set of neighborhoods and alternate them. This approach is the basis for VNS algorithms. The idea of alternating neighborhoods is easy to adapt to various problems [76–78] and highly efficient, which makes it very useful for solving NP-hard problems including clustering, location, and vehicle routing problems. In [111,112], Brimberg and Mladenovic and Miskovic et al. used the VNS for solving various facility location problems. Cranic et al. [113] as well as Hansen and Mladenovic [114] proposed and developed a parallel VNS algorithm for the k-median problem. In [115], a VNS algorithm was used for a vehicle routing and driver scheduling problems by Wen et al.

The ways of neighborhood alternation may differ significantly. Many VNS algorithms are not even classified by their authors as VNS algorithms. For example, the algorithm in [57] alternates between discrete and continuous problems: when solving a discrete problem, the set of local optima is replenished, and then such local optima are chosen as elements of the initial solution of the continuous problem. A similar idea of the recombinator k-means algorithm was proposed by C. Baldassi [116]. This algorithm restarts the k-means procedure, using the results of previous runs as a reservoir of candidates for the new initial solutions, exploiting the popular k-means++ seeding

algorithm to piece them together into new, promising initial configurations. Thus, the k-means search alternates with the discrete problem of finding an optimal initial centroid combination.

VNS class includes a very efficient abovementioned J-Means algorithm [80], which alternates search in a SWAP neighborhood and the use of Lloyd's procedure. Even when searching only in the $SWAP_1$ neighborhood, the J-Means results can be many times better than the results of Lloyd's procedure launched in the multi-start mode, as shown in [62,97].

In [50], Kochetov et al. describe such basic schemes of VNS algorithms as variable neighborhood descent (VND, see Algorithm 4) [117] and randomized Variable Neighborhood Search (RVNS, see Algorithm 5) [50].

---

**Algorithm 4.** *VND(S)*

---

**Require:** Initial solution $S$, selected neighborhoods $n_l$, $l = \left\{\overline{1, l_{max}}\right\}$.
**repeat**
> $l \leftarrow 1$;
> **while**$l \leq l_{max}$**do**
> > search for $S' \in n_l(S) : f(S') = \min\{f(Y)|Y \in n_l(S)\}$;
> > **if** $f(S') < f(S)$ **then** $S \leftarrow S'; l \leftarrow 1$ **else** $l \leftarrow l+1$ **end if**;
> **end while**;

**until** the stop conditions are satisfied.

---

---

**Algorithm 5.** *RVNS(S)*

---

**Require:** Initial solution $S$, selected neighborhoods $n_l$, $l = \left\{\overline{1, l_{max}}\right\}$.
**repeat**
> $l \leftarrow 1$;
> **While** $l \leq l_{max}$ **do**
> > select randomly $S' \in n_l(S)$;
> > **if** $f(S') < f(S)$ **then** $S \leftarrow S'; l \leftarrow 1$ **else** $l \leftarrow l+1$ **end if**;
> **end while**;

**until** the stop conditions are satisfied.

---

Algorithms of the RVNS scheme are more efficient when solving large-scale problems [50], when the use of deterministic VND requires too large computational costs per each iteration. In many efficient algorithms, $l_{max} = 2$. For example, the J-Means algorithm combines a SWAP neighborhood search with Lloyd's procedure.

As a rule, algorithm developers propose to move from neighborhoods of lower cardinality to wider neighborhoods. For instance, in [79], the authors propose a sequential search in the neighborhoods $GREEDY_1 \rightarrow GREEDY_{random} \rightarrow GREEDY_k \rightarrow GREEDY_1 \rightarrow \ldots$ Here, $GREEDY_{random}$ is a neighborhood with randomly selected $r \in \left\{\overline{2, k-1}\right\}$. In this case, the initial neighborhood type has a strong influence on the result [79]. However, the best initial value of parameter $r$ is hardly predictable.

In this article, we propose a new RVNS algorithm which involves $GREEDY_r$ neighborhood search with a gradually decreasing $r$ and automatic adjustment of the initial $r$ value. Computational experiments show the advantages of this algorithm in comparison with the algorithms searching in SWAP neighborhoods as well as in comparison with known search algorithms with $GREEDY_r$ neighborhoods.

*2.5. New Algorithm*

A search in a $GREEDY_r$ neighborhood with a fixed $r$ values, on various practical problems listed in the repositories [108,109,118], shows that the result (the value of the objective function) essentially depends on $r$, and this dependence differs for various problems, even if the problems have similar basic numerical characteristics, such as the number of data vectors $N$, their dimension $d$, and the number of clusters $k$. The results are shown on Figures 2 and 3. At the same time, our experiments show that at the first iterations, the use of Algorithm 3 almost always leads to an improvement in the *SSE* value,

and then the probability of such a success decreases. Moreover, the search in neighborhoods with large $r$ values stops giving improving results sooner, while the search in neighborhoods with small $r$, in particular, with $r = 1$, enables us to obtain the improved solutions during a longer time. The search in the GREEDY$_1$ neighborhood corresponds to the adjustment of individual centroid positions. Thus, the possible decrement of the objective function value is not the same for different values of $r$.

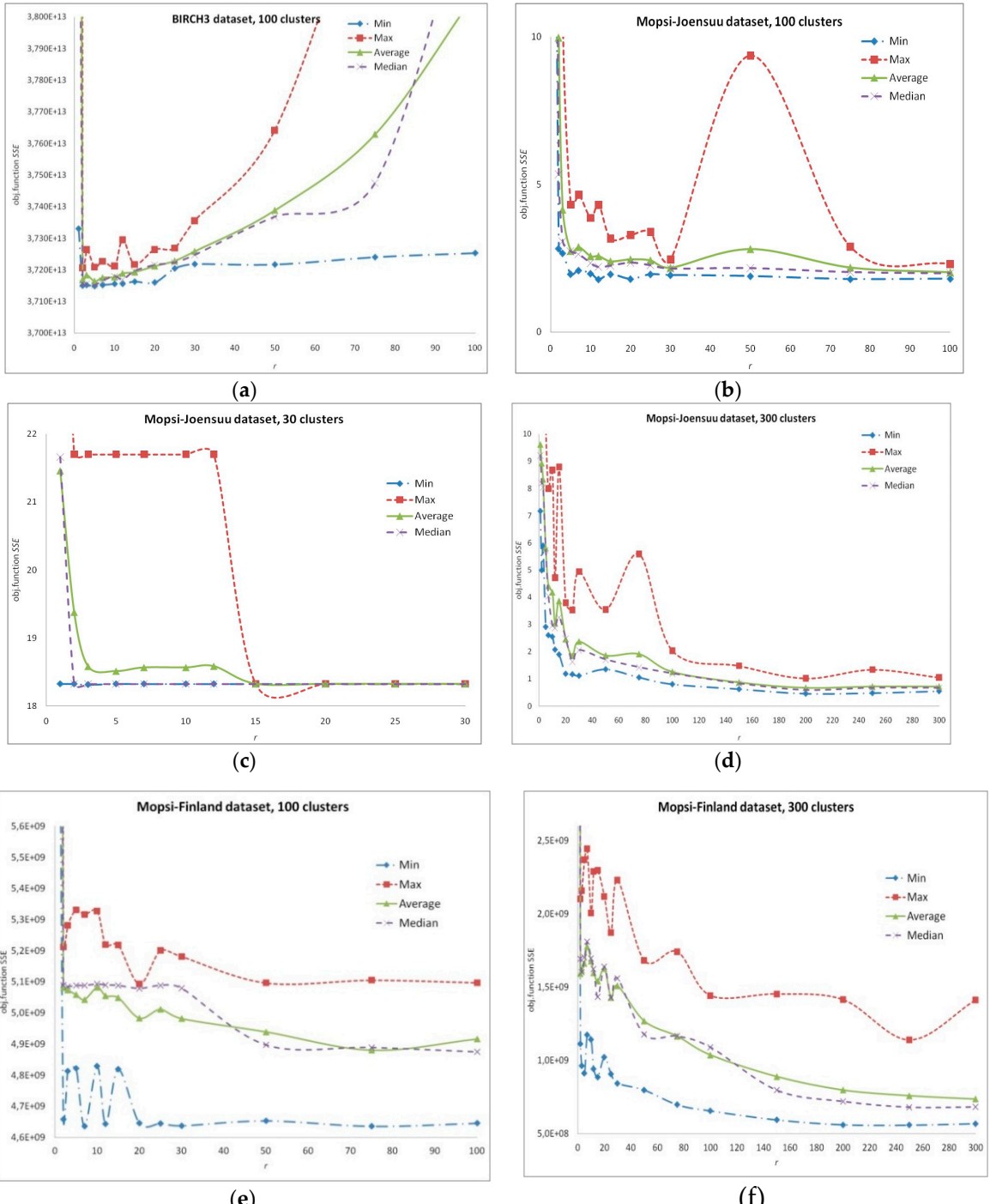

**Figure 2.** Search in GREEDY$r$ neighborhoods. Dependence of the result on $r$: (**a**) BIRCH3 data set, 100 clusters, $10^5$ data vectors, time limitation 10 s; (**b–d**) Mopsi-Joensuu data set, 30, 100 and 300 clusters, 6014 data vectors, time limitation 5 s; (**e–f**) Mopsi-Finland data set, 100 and 300 clusters, 13,467 data vectors, time limitation 5 s.

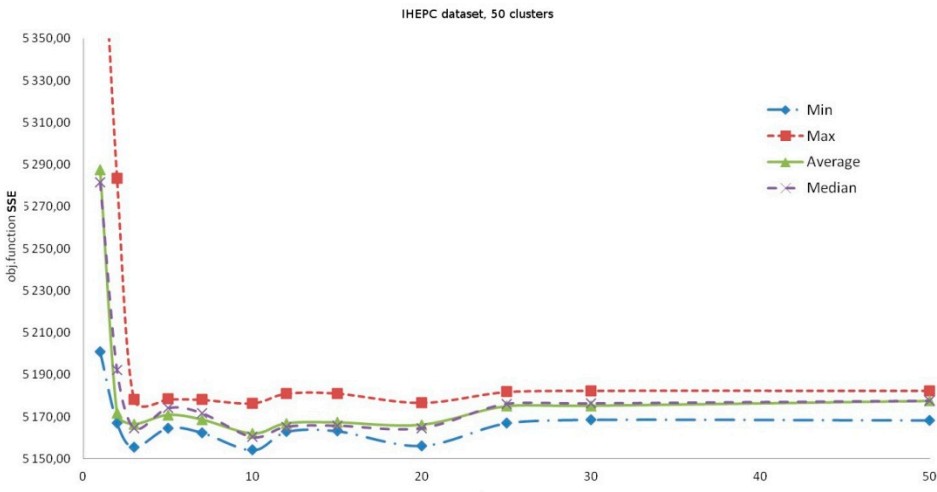

**Figure 3.** Search in GREEDY*r* neighborhoods. Dependence of the result on *r*: Individual Household Electric Power Consumption (IHEPC) data set, 50 clusters, 2,075,259 data vectors, time limitation 5 min.

We propose the following sequence of neighborhoods: $GREEDY_{r0} \rightarrow GREEDY_{r1} \rightarrow GREEDY_{r2} \rightarrow \ldots \rightarrow GREEDY_1 \rightarrow GREEDY_k \rightarrow \ldots$. Here, *r* values gradually decrease: $r0 > r1 > r2 \ldots$. After reaching $r = 1$, the search continues in the $GREEDY_k$ neighborhood, and after that the value of *r* starts decreasing again. Moreover, the *r* value fluctuates within certain limits at each stage of the search.

This algorithm can be described as follows (Algortithm 6).

---

**Algorithm 6.** *DecreaseGreedySearch(S,r₀)*

---

**Require:** Initial solution *S*, initial $r = r_0 \in \overline{\{1,k\}}$.
select randomly $S_2 \subset \{A_1, \ldots, A_N\}$, $|S_2| = k$; $S_2 \leftarrow Lloyd(S_2)$;
**repeat**
        $n_{repeats} \leftarrow \max\{1, [k/r]\}$;
        **for** $i = \overline{1, n_{repeats}}$ **do**
            1. select randomly $r' \in \left\{ \overline{max\{1, \left[\frac{r_0}{2}\right]\}, r_0} \right\}$;
            2. $S' \leftarrow Greedy(S, S_2, r')$;
            3. **if** $SSE(S') < SSE(S)$ **then** $S \leftarrow S'$ **end if**;
        **endfor**;
select randomly $S_2 \subset \{A_1, \ldots, A_N\}$, $|S_2| = k$; $S_2 \leftarrow Lloyd(S_2)$;
        **if** Steps 1–3 have not changed *S*
        **then**
            **if** $r = 1$ **then** $r_0 \leftarrow k$ **else** $r_0 \leftarrow max\{1, \left[\frac{r}{2}\right] - 1\}$ **end if**;
        **end if**;
        **until** the stop conditions are satisfied (time limitation).

---

Genetic algorithms with greedy agglomerative heuristics are known to perform better than VNS algorithms with sufficient computation time [79,90] which results in better *SSE* values. Despite this, the limited time and computational complexity of the *Greedy*() procedure as a genetic crossover operator leads to a situation when genetic algorithms may have enough time to complete a very limited number of crossover operations and often only reach the second or third generation of solutions. Under these conditions, VNS algorithms are a reasonable compromise of the computation cost and accuracy.

The choice of the initial value of parameter $r_0$ is highly important. Such a choice is quite simply carried out by a reconnaissance search with different $r_0$ values. The algorithm with such an automatic adjustment of the parameter $r_0$ by performing a reconnaissance search is described as follows (Algorithm 7).

---

**Algorithm 7.** *AdaptiveGreedy* (*S*) solver

---

**Require:** the number of reconnaissance search iterations $n_{recon}$.
select randomly $S \subset \{A_1, \ldots, A_N\}$, $|S| = k$; $S \leftarrow Lloyd(S)$;
**for** $i = \overline{1, n_{recon}}$ **do**
        select randomly $S_i \subset \{A_1, \ldots, A_N\}$, $|S_i| = k$; $S_i \leftarrow Lloyd(S_i)$;
**end for;**
$r \leftarrow k$;
**repeat**
        $S_r'' \leftarrow S$; $n_{repeats} \leftarrow \max\{1, [k/r]\}$;
        **for** $i = \overline{1, n_{recon}}$ **do**
            **for** $i = \overline{1, n_{repeats}}$ **do**
                $S' \leftarrow Greedy(S_r'', S_i, r)$; **if** $SSE(S') < SSE(S_r'')$ **then** $S_r'' \leftarrow S'$ **end if;**
            **end for;**
            **end for;**
            $r \leftarrow max\{1, \left[\frac{r}{2}\right] - 1\}$;
**until** $r = 1$ ;
select the value $r$ with minimum value of $SSE(S_r'')$;
$r_0 \leftarrow \min\{1.5r, k\}$ ;
*DecreaseGreedySearch*($S_r''$, $r_0$).

---

Results of computational experiments described in the next Section show that our new algorithm, which sequentially decreases the value of the parameter $r_0$, has an advantage over the known VNS algorithms.

## 2.6. CUDA Implementation

The greedy agglomerative procedure (*BasicGreedy*) is computationally expensive. In Algorithm 2, the objective function calculation $F_{i'} \leftarrow SSE(S \backslash \{X_i\})$ is performed more than $(K - k) \cdot k$ times in each iteration, and after that, *Lloyd*() procedure is executed. Therefore, such algorithms are traditionally considered as methods for solving comparatively small problems (hundreds of thousands of data points and hundreds of clusters). However, the rapid development of the massive parallel processing systems (GPUs) enables us to solve the large-scale problems with reasonable time expenses (seconds). Parallel (CUDA) implementation of the algorithms for the *Lloyd*() procedure is known [101,102], and we used this approach in our experiments.

Graphic processing units (GPUs) accelerate computations with the use of multi-core computing architecture. The CUDA (compute unified device architecture) is the most popular programming platform which enables us to use general-purpose programming languages (e.g., C++) for compiling GPU programs. The programming model uses the single instruction multiple thread (SIMT) principle [119]. We can declare a function in the CUDA program a "kernel" function and run this function on the steaming multiprocessors. The threads are divided into blocks. Several instances of a kernel function are executed in parallel on different nodes (blocks) of a computation grid. Each thread can be identified by special *threadIdx* variable. Each thread block is identified by *blockIdx* variable. The number of threads in a block is identified by *blockDim* variable. All these variables are 3-dimensional vectors (dimensions $x$, $y$, $z$). Depending on the problem solved, the interpretation of these dimensions may differ. For processing 2D graphical data, $x$ and $y$ are used for identifying pixel coordinates.

The most computationally expensive part of Lloyd's procedure is distance computation and comparison (Step 1 of Algorithm 1). This step can be seamlessly parallelized if we calculate distances from each individual data vector in a separate thread. Thus, *threadIdx.x* and *blockIdx.x* must indicate a data vector. The same kernel function prepares data needed for centroid calculation (Step 2 of Algorithm 1). Such data are the sum of data vector coordinates in a specific cluster $sum_j = \sum_{i \in \{\overline{1,N}\}: A_i \in C_j} A_i$ and the cardinality of the cluster $counter_j = |C_j|$. Here, $j$ is the cluster number. Variable $sum_j$ is a vector (1-dimensional array in program realization).

To perform Step 1 of Algorithm 1 on a GPU, after initialization $sum_j \leftarrow 0$ and $counter_j \leftarrow 0$, the following procedure (Algorithm 8) runs on $(N + blockDim.x) / blockDim.x$ nodes of computation grid, with $blockDim.x$ threads in each block (in our experiments, $blockDim.x = 512$):

---

**Algorithm 8.** *CUDA kernel implementation of Step 1 in Lloyd's procedure (Algorithm 1)*

---

$i \leftarrow blockIdx.x \cdot blockDim.x + threadIdx.x$;
**if** $i > N$ **then** return **end if**;
$D_{nearest} \leftarrow +\infty$; // *distance from $A_i$ to the nearest centroid*
**for** $j = \overline{1,k}$ **do**
    **if** $\|A_j - X_i\| < D_{nearest}$ **then**
        $D_{nearest} \leftarrow A_j - X_i$;
        $n \leftarrow j$;
    **end if**
**end for**;
$sum_n \leftarrow sum_n + A_n$;
$counter_n \leftarrow counter_n + 1$;
$SSE \leftarrow SSE + D_{nearest}^2$ . // *objective function adder*

---

If $sum_j$ and $counter_j$ are pre-calculated for each cluster then Step 2 of Algorithm 1 is reduced to a single arithmetic operation for each cluster: $X_j = sum_j/counter_j$. If the number of clusters is not huge, this operation does not take significant computation resources. Nevertheless, its parallel implementation is even simpler: we organize $k$ treads, and each thread calculates $X_j$ for an individual cluster. Outside Lloyd's procedure, we use Algorithm 8 for SSE value estimation (variable $SSE$ must be initialized by 0 in advance).

The second computationally expensive part of the *BasicGreegy*() algorithm is estimation of the objective function value after eliminating a centroid [120]: $F_{i'} = SSE(S\backslash\{X_i\})$. Having calculated $SSE(S)$, we may calculate as $SSE(S\backslash\{X_i\})$ as

$$F_{i'} = SSE(S\backslash\{X_i\}) = SSE(S) + \sum_{l=1}^{N} \Delta D_l \tag{5}$$

where

$$\Delta D_l = \begin{cases} 0, & A_l \notin C_i, \\ \left(min_{j \in \{\overline{1,k}\}, \, j \neq i} \|X_j - A_l\|\right)^2 - \|X_i - A_l\|^2, & A_l \in C_i. \end{cases}$$

For calculating (5) on a GPU, after initializing $F_i \leftarrow SSE(S)$, the following kernel function (Algorithm 9) runs for each data vector.

---

**Algorithm 9.** *CUDA kernel implementation of calculating $F_i \leftarrow SSE(S\backslash\{X_i\})$ in BasicGreedy procedure (Algorithm 2)*

---

**Require:** index $i$ of centroid being eliminated.
$l \leftarrow blockIdx.x \cdot blockDim.x + threadIdx.x$;
**if** $l > N$ **then** return **end if**;
$D_{nearest} \leftarrow +\infty$; // *distance from $A_l$ to the nearest centroid except $X_i$*
**for** $j = \overline{1,k}$ **do**
    **if** $l \neq i$ **and** $A_j - X_i < D_{nearest}$ **then**
        $D_{nearest} \leftarrow \|A_j - X_i\|$;
    **end if**
**end for**;
$F_i \leftarrow F_i + D_{nearest}^2 - \|X_i - A_l\|^2$;

---

All distance calculations for $GREEDY_r$ neighborhood search are performed by Algorithms 8 and 9. A similar kernel function was used for accelerating the local search in *SWAP* neighborhoods. In this function, after eliminating a centroid, a data point is included in solution *S* as a new centroid.

All other parts of new and known algorithms were implemented on the CPU.

*2.7. Benchmarking Data*

In all our experiments, we used the classic data sets from the UCI Machine Learning and Clustering basic benchmark repositories [108,109,118]:

(a) Individual household electric power consumption (IHEPC)—energy consumption data of households during several years (more than 2 million data vectors, 7 dimensions), 0–1 normalized data, "date" and "time" columns removed;
(b) BIRCH3 [121]: one hundred of groups of points of random size on a plane ($10^5$ data vectors, 2 dimensions);
(c) S1 data set: Gaussian clusters with cluster overlap (5000 data vectors, 2 dimensions);
(d) Mopsi-Joensuu: geographic locations of users (6014 data vectors, 2 dimensions) in Joensuu city;
(e) Mopsi-Finland: geographic locations of users (13,467 data vectors, 2 dimensions) in Finland.

Mopsi-Joensuu and Mopsi-Finland are "geographic" data sets with a complex cluster structure, formed under the influence of natural factors such as the geometry of the city, transport communications, and urban infrastructure (Figure 4).

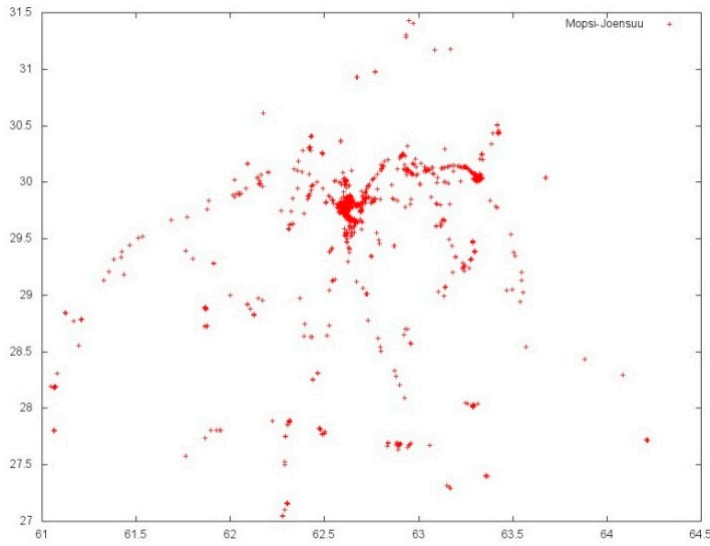

**Figure 4.** Mopsi-Joensuu data set visualization.

In our study, we do not take into account the true labeling provided by the data set (if it is known), i.e., the given predictions for known classes, and focus on the minimization of SSE only.

*2.8. Computational Environment*

For our computational experiments, we used the following test system: Intel Core 2 Duo E8400 CPU, 16GB RAM, NVIDIA GeForce GTX1050ti GPU with 4096 MB RAM, floating-point performance 2138 GFLOPS. This choice of the GPU hardware was made due to its prevalence, and also one of the best values of the price/performance ratio. The program code was written in C++. We used Visual C++ 2017 compiler embedded into Visual Studio v.15.9.5, NVIDIA CUDA 10.0 Wizards, and NVIDIA Nsight Visual Studio Edition CUDA Support v.6.0.0.

## 3. Results

For all data sets, 30 attempts were made to run each of the algorithms (see Tables 1 and A1, Tables A2–A11 in Appendix A).

**Table 1.** Comparative results for all data sets (best of known algorithms vs. new algorithm).

| Algorithm or Neighborhood | Achieved SSE Summarized After 30 Runs | | | | | p-Values and Statistical Significance of Difference in Results |
|---|---|---|---|---|---|---|
| | Min (Record) | Max (Worst) | Average | Median | Std.dev | |
| BIRCH3 data set. $10^5$ data vectors in $\mathbb{R}^2$, k = 300 clusters, time limitation 10 s | | | | | | |
| GREEDY$_{200}$ | $1.30773 \times 10^{13}$ | $1.31172 \times 10^{13}$ | $1.30916 \times 10^{13}$ | $1.30912 \times 10^{13}$ | $1.08001 \times 10^{10}$ | $p_t = 0.4098 \leftrightarrow$ |
| AdaptiveGreedy | $1.30807 \times 10^{13}$ | $1.31113 \times 10^{13}$ | $1.30922 \times 10^{13}$ | $1.30925 \times 10^{13}$ | $0.87731 \times 10^{10}$ | $p_U = 0.2337 \Leftrightarrow$ |
| BIRCH3 data set. $10^5$ data vectors in $\mathbb{R}^2$, k = 100 clusters, time limitation 10 s | | | | | | |
| GREEDY$_5$ | $3.71485 \times 10^{13}$ | $3.72087 \times 10^{13}$ | $3.71644 \times 10^{13}$ | $3.71518 \times 10^{13}$ | $2.22600 \times 10^{10}$ | $p_t = 0.0701 \leftrightarrow$ |
| AdaptiveGreedy | $3.71484 \times 10^{13}$ | $3.72011 \times 10^{13}$ | $3.71726 \times 10^{13}$ | $3.71749 \times 10^{13}$ | $2.02784 \times 10^{10}$ | $p_U = 0.1357 \Leftrightarrow$ |
| Mopsi-Joensuu data set. 6014 data vectors in $\mathbb{R}^2$, k = 300 clusters, time limitation 5 s | | | | | | |
| GH-VNS3 | 0.4321 | 0.6838 | 0.6024 | 0.6139 | 0.0836 | $p_U = 0.00005 \Uparrow$ |
| GREEDY$_{200}$ | 0.4555 | 1.0154 | 0.6746 | 0.5882 | 0.2163 | $p_t < 0.00001 \uparrow$ |
| AdaptiveGreedy | 0.3128 | 0.6352 | 0.4672 | 0.4604 | 0.1026 | |
| Mopsi-Joensuu data set. 6014 data vectors in $\mathbb{R}^2$, k = 100 clusters, time limitation 5 s | | | | | | |
| GREEDY$_{100}$ | 1.8021 | 2.2942 | 2.0158 | 1.9849 | 0.1860 | $p_t = 0.0910 \leftrightarrow$ |
| GH-VNS3 | 1.7643 | 2.7357 | 2.0513 | 1.9822 | 0.2699 | $p_U = 0.0042 \Uparrow$ |
| AdaptiveGreedy | 1.7759 | 2.3265 | 1.9578 | 1.9229 | 0.1523 | |
| Mopsi-Joensuu data set. 6014 data vectors in $\mathbb{R}^2$, k = 30 clusters, time limitation 5 s | | | | | | |
| GH-VNS1 | 18.3147 | 18.3255 | 18.3238 | 18.3253 | 0.0039 | $p_t = 0.4118 \leftrightarrow$ |
| AdaptiveGreedy | 18.3146 | 18.3258 | 18.3240 | 18.3253 | 0.0037 | $p_U = 0.2843 \Leftrightarrow$ |
| Mopsi- Finland data set.13,467 data vectors in $\mathbb{R}^2$, k = 300 clusters, time limitation 5 s | | | | | | |
| GH-VNS3 | $5.33373 \times 10^8$ | $7.29800 \times 10^8$ | $5.74914 \times 10^8$ | $5.48427 \times 10^8$ | $5.05346 \times 10^7$ | $p_t = 0.1392 \leftrightarrow$ |
| AdaptiveGreedy | $5.27254 \times 10^8$ | $7.09410 \times 10^8$ | $5.60867 \times 10^8$ | $5.38952 \times 10^8$ | $4.89257 \times 10^7$ | $p_U = 0.0049 \Uparrow$ |
| Mopsi-Finland data set. 13,467 data vectors in $\mathbb{R}^2$, k = 30 clusters, time limitation 5 s | | | | | | |
| GH-VNS3 | $3.42528 \times 10^{10}$ | $3.47955 \times 10^{10}$ | $3.43826 \times 10^{10}$ | $3.43474 \times 10^{10}$ | $1.02356 \times 10^8$ | $p_t = 0.0520 \leftrightarrow$ |
| AdaptiveGreedy | $3.42528 \times 10^{10}$ | $3.47353 \times 10^{10}$ | $3.43385 \times 10^{10}$ | $3.43473 \times 10^{10}$ | $1.03984 \times 10^8$ | $p_U = 0.0001 \Uparrow$ |
| S1 data set. 5000 data vectors in $\mathbb{R}^2$, k = 15 clusters, time limitation 1 second | | | | | | |
| GH-VNS2 | $8.91703 \times 10^{12}$ | $8.91703 \times 10^{12}$ | $8.91703 \times 10^{12}$ | $8.91703 \times 10^{12}$ | 0.0000 | $p_t = 0.5 \leftrightarrow$ |
| AdaptiveGreedy | $8.91703 \times 10^{12}$ | $8.91703 \times 10^{12}$ | $8.91703 \times 10^{12}$ | $8.91703 \times 10^{12}$ | 0.0000 | $p_U = 0.5 \Leftrightarrow$ |
| S1 data set. 5000 data vectors in $\mathbb{R}^2$, k = 50 clusters, time limitation 1 second | | | | | | |
| GH-VNS1 | $3.74310 \times 10^{12}$ | $3.76674 \times 10^{12}$ | $3.74911 \times 10^{12}$ | $3.74580 \times 10^{12}$ | $6.99859 \times 10^9$ | $p_t = 0.3571 \leftrightarrow$ |
| AdaptiveGreedy | $3.74340 \times 10^{12}$ | $3.76313 \times 10^{12}$ | $3.74851 \times 10^{12}$ | $3.75037 \times 10^{12}$ | $5.56298 \times 10^9$ | $p_U = 0.28434 \Leftrightarrow$ |
| IHEPC data set. 2,075,259 data vectors in $\mathbb{R}^7$, k = 50 clusters, time limitation 5 min | | | | | | |
| GREEDY$_{10}$ | 5154.2017 | 5176.4502 | 5162.0460 | 5160.4014 | 7.2029 | $p_t = 0.008 \uparrow$ |
| AdaptiveGreedy | 5153.5640 | 5163.9316 | 5157.0822 | 5155.5198 | 3.6034 | $p_U = 0.001 \Uparrow$ |

Note: "↑", "⇑": the advantage of the new algorithms over known algorithms is statistically significant ("↑" for *t*-test and "⇑" for Mann–Whitney U test), "↓", "⇓": the disadvantage of the new algorithm over known algorithms is statistically significant; "↔", "⇔": the advantage or disadvantage is statistically insignificant. Significance level is 0.01.

For comparison, we ran local search in various GREEDY*r* neighborhoods at fixed *r* value. In addition, we ran various known Variable Neighborhood Search (VNS) algorithms with GREEDY*r* neighborhoods [79], see algorithms GH-VNS1-3. These algorithms use the same sequence of neighborhood types (GREEDY$_1 \rightarrow$GREEDY$_{random} \rightarrow$GREEDY$_k$) and differ in the initial neighborhood type: GREEDY$_1$ for GH-VNS1, GREEDY$_{random}$ for GH-VNS2, and GREEDY$_k$ GH-VNS3. Unlike our new *AdaptiveGreedy*() algorithm, GH-VNS1-3 algorithms increase *r* values, and this increase is not gradual. In addition, we included the genetic algorithm (denoted "GA-1" in Tables A1–A11) with the single-point crossover [103], real-valued genes encoded by centroid positions, and the uniform random mutation (probability 0.01). For algorithms launched in the multi-start mode (j-Means algorithm and

Lloyd's procedure), only the best results achieved in each attempt were recorded. In Tables A1–A11, such algorithms are denoted Lloyd-MS and j-Means-MS, respectively.

The minimum, maximum, average, and median objective function values and its standard deviation were summarized after 30 runs. For all algorithms, we used the same realization of the *Lloyd*() procedure which consume the absolute majority of the computation time.

The best average and median values of the objective function (1) are underlined. We compared the new *AdaptiveGreedy*() algorithm with the known algorithm which demonstrated the best median and average results (Table 1). For comparison, we used the *t*-test [122,123] and non-parametric Wilcoxon-Mann-Whitney U test (Wilcoxon rank sum test) [124,125] with *z* approximation.

To compare the results obtained by our new algorithm, we tested the single-tailed null hypothesis $H_0$: $SSE_{AdaptiveGreedy} = SSE_{known}$ (the difference in the results is statistically insignificant) and the research hypothesis $H_1$: $SSE_{AdaptiveGreedy} < SSE_{known}$ (statistically different results, the new algorithm has an advantage). Here, $SSE_{AdaptiveGreedy}$ are results ontained by *AdaptiveGreedy*() algorithm, $SSE_{known}$ are results of the best-known algorithm. For *t*-test comparison, we selected the algorithm lowest in average SSE value, and for Wilcoxon–Mann–Whitney U test comparison, we selected the algorithm with the lowest SSE median value. For both tests, we calculated the *p*-values (probability of the null-hypothesis acceptance), see $p_t$ for the *t*-test and $p_u$ for the Wilcoxon–Mann–Whitney U test in Table 1, respectively. At the selected significance level $p_{sig} = 0.01$, the null hypothesis is accepted if $p_t > 0.01$ or $p_U > 0.01$. Otherwise, the difference in algorithm results should be considered statistically significant. If the null hypothesis was accepted, we also tested a pair of single-tailed hypotheses $SSE_{AdaptiveGreedy} = SSE_{known}$ and $SSE_{AdaptiveGreedy} > SSE_{known}$.

In some cases, the Wilcoxon–Mann–Whitney test shows the statistical significance of the differences in results, while the *t*-test does not confirm the benefits of the new algorithm. Figure 5 illustrates such a situation. Both algorithms demonstrate approximately the same results. Both algorithms periodically produce results that are far from the best SSE values, which is expressed in a sufficiently large value of the standard deviation. However, the results of the new algorithm are often slightly better, which is confirmed by the rank test.

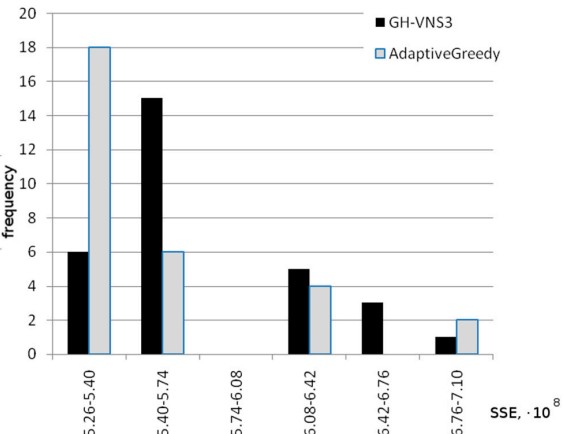

**Figure 5.** Frequency diagram of the results (our new algorithm vs. the best of other tested algorithms, GH-VNS3), Mopsi-Finland data set, 300 clusters, 13,467 data vectors, time limitation 5 s, 30 runs of each algorithm.

In the comparative analysis of algorithm efficiency, the choice of the unit of time plays an important role. The astronomical time spent by an algorithm strongly depends on its implementation, the ability of the compiler to optimize the program code, and the fitness of the hardware to execute the code of a specific algorithm. Algorithms are often estimated by comparing the number of iterations performed (for example, the number of population generations for a GA) or the number of evaluations of the objective function.

However, the time consumption for a single iteration of a local search algorithm depends on the neighborhood type and number of elements in the neighborhood, and this dependence can be exponential. Therefore, comparing the number of iterations is unacceptable. Comparison of the objective function calculations is also not quite correct. Firstly, the *Lloyd*() procedure which consumes almost all of the processor time, does not calculate the objective function (1) directly. Secondly, during the operation of the greedy agglomerative procedure, the number of centroids changes (decreases from $k + r$ down to $k$), and the time spent on computing the objective function also varies. Therefore, we nevertheless chose astronomical time as a scale for comparing algorithms. Moreover, all the algorithms use the same implementation of the *Lloyd*() algorithm launched under the same conditions.

In our computational experiments, the time limitation was used as the stop condition for all algorithms. For all data sets except the largest one, we have chosen a reasonable time limit to use the new algorithm in interactive modes. For IHEPC data and 50 clusters, a single run of the *BasicGreedy*() algorithm on the specified hardware took approximately 0.05 to 0.5 s. It is impossible to evaluate the comparative efficiency of the new algorithm in several iterations, since in this case, it does not have enough time to change the neighborhood parameter *r* at least once. We have increased the time to a few minutes. This time limit does not correspond to modern concepts of interactive modes of operation. Nevertheless, the rapid development of parallel computing requires the early creation of efficient algorithmic schemes. Our experiments were performed on a mass-market system. Advanced systems may cope with such large problems much faster.

As can be seen from Figure 6, the result of each algorithm depends on the elapsed time. Nevertheless, an advantage of the new algorithm is evident regardless of the chosen time limit.

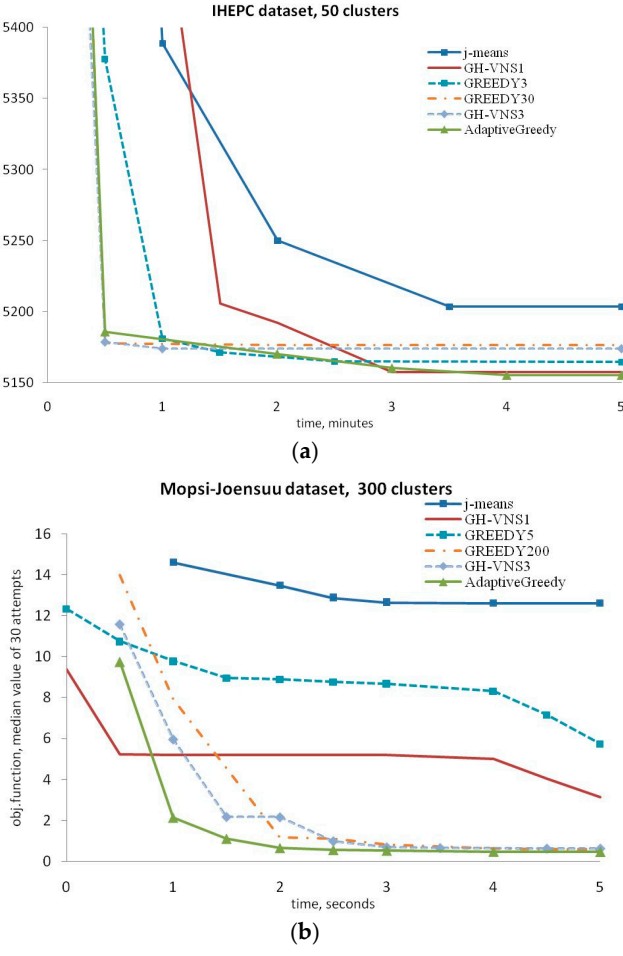

**Figure 6.** *Cont.*

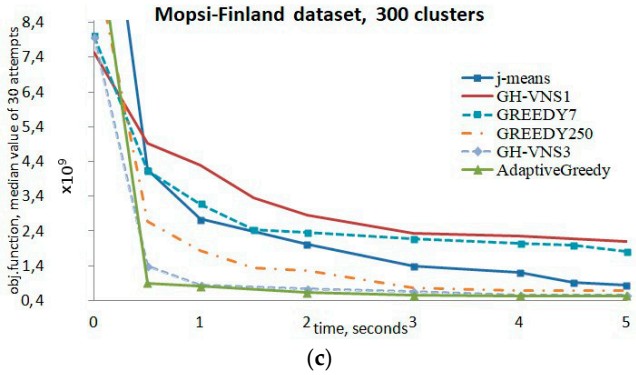

(c)

**Figure 6.** Comparative analysis of the convergence speed. Dependence of the median result on computation time for: (**a**) Individual Household Electric Power Consumption (IHEPC) data set, 50 clusters, 2,075,259 data vectors, time limitation 5 min; (**b**) Mopsi-Joensuu data set, 300 clusters, 6014 data vectors, time limitation 5 s; (**c**) Mopsi-Finland data set, 300clusters, 13,467 data vectors, time limitation 5 s.

To test the scalability of the proposed approach and the efficiency of the new algorithm on other hardware, we carried out additional experiments with NVIDIA GeForce 9600GT GPU, 2048 MB RAM, 336 GFLOPS. The declared performance of this simpler equipment is approximately 6 times lower. The results of experiments with proportional increase of time limitation are shown in Table 2. The difference with the results in Table 1 is obviously insignificant.

**Table 2.** Additional benchmarking on NVIDIA GeForce 9600GT GPU. Comparative results for Mopsi-Finland data set.13,467 data vectors in $\mathbb{R}^2$, time limitation 30 s.

| Algorithm or Neighborhood | Achieved SSE Summarized After 30 Runs | | | | |
|---|---|---|---|---|---|
| | **Min (Record)** | **Max (Worst)** | **Average** | **Median** | **Std.dev** |
| $k = 300$ | | | | | |
| GH-VNS3 | $5.33373 \times 10^8$ | $7.29800 \times 10^8$ | $5.85377 \times 10^8$ | $5.52320 \times 10^8$ | $5.59987 \times 10^7$ |
| AdaptiveGreedy | $5.27254 \times 10^8$ | $7.09410 \times 10^8$ | $5.59033 \times 10^8$ | $5.38888 \times 10^8$ | $4.60585 \times 10^7$ |
| $k = 30$ | | | | | |
| GH-VNS2 | $3.42528 \times 10^{10}$ | $3.48723 \times 10^{10}$ | $3.43916 \times 10^{10}$ | $3.43474 \times 10^{10}$ | $1.46818 \times 10^8$ |
| GH-VNS3 | $3.42528 \times 10^{10}$ | $3.46408 \times 10^{10}$ | $3.43731 \times 10^{10}$ | $3.43474 \times 10^{10}$ | $7.81989 \times 10^7$ |
| AdaptiveGreedy | $3.42528 \times 10^{10}$ | $3.46274 \times 10^{10}$ | $3.43337 \times 10^{10}$ | $3.43473 \times 10^{10}$ | $8.13882 \times 10^7$ |

The ranges of SSE values in the majority of Tables A1–A11 are narrow, nevertheless, the differences are statistically significant in several cases, see Table 1. In all cases, our new algorithm outperforms known ones or demonstrates approximately the same efficiency (difference in the results is statistically insignificant). Moreover, the new algorithm demonstrates the stability of its results (narrow range of objective function values).

Search results in both $SWAP_r$ and $GREEDY_r$ neighborhoods depend on a correct choice of parameter $r$ (the number of replaced or added centroids). However, in general, local search algorithms with $GREEDY_r$ neighborhoods outperform the $SWAP_r$ neighborhood search. A simple reconnaissance search procedure enables the further improvement of the efficiency.

## 4. Discussion

The advantages of our algorithm are statistically significant for a large problem (IHEPC data), as well as for problems with a complex data structure (Mopsi-Joensuu and Mopsi-Finland data). The Mopsi data sets contain geographic coordinates of Mopsi users, which are extremely unevenly distributed in accordance with the natural organization of the urban environment, depending on street directions and urban infrastructure (Figure 6). In this case, the aim of clustering is to find some natural

groups of users according to a geometric/geographic principle for assigning them to $k$ service centers (hubs) such as shopping centers, bus stops, wireless network base stations, etc.

Often, geographical data sets show such a disadvantage of Lloyd's procedure as its inability to find a solution close to the exact one. Often, on such data, the value of the objective function found by the Lloyd's procedure in the multi-start mode turns out to be many times greater than the values obtained by other algorithms, such as J-Means or RVNS algorithms with SWAP neighborhoods. As can be seen from Tables A2, A3 and A5 in Appendix A, for such data, GREEDY$_r$ neighborhoods search provides significant advantages within a limited time, and our new self-adjusting *AdaptiveGreedy*() solver enhances these advantages.

The VNS algorithmic framework is useful for creating effective computational tools intended to solve complex practical problems. Embedding the most efficient types of neighborhoods in this framework depends on the problem type being solved. In problems such as k-means, the search in neighborhoods with specific parameters strongly depends not only on the generalized numerical parameters of the problems, such as the number of clusters, number of data vectors, and the search space dimensionality, but also on the internal data structure. In general, the comparative efficiency of the search in GREEDY$_r$ neighborhoods for certain types of practical problems and for specific data sets remains an open question. Nevertheless, the algorithm presented in this work, which automatically performs the adjustment of the most important parameter of such neighborhoods, enables its user to obtain the best result which the variable neighborhood search in GREEDY$_r$ is able to provide, without preliminary experiments in all possible GREEDY$_r$ neighborhoods. Thus, the new algorithm is a more versatile computational tool in comparison with the known VNS algorithms.

Greedy agglomerative procedures are widely used as crossover operators in genetic algorithms [46,88,90,110]. In this case, most often, the "parent" solutions are merged completely to obtain an intermediate solution with an excessive number of centers or centroids [46,88], which corresponds to the search in the GREEDY$_k$ neighborhood (one of the crossed "parent" solutions acts as the parameter $S_2$), although, other versions of the greedy agglomerative crossover operator are also possible [90,110]. Such algorithms successfully compete with the advanced local search algorithms discussed in this article.

Self-configuring evolutionary algorithms [126–128] have been widely used for solving various optimization problems. An important direction of the further research is to study the possibility of adjusting the parameter $r$ in greedy agglomerative crossover operators of genetic algorithms. Such procedures with self-adjusting parameter $r$ could lead to a further increase in the accuracy of solving the $k$-means problem with respect to the achieved value of the objective function. Such evolutionary algorithms could also involve a reconnaissance search, which would then continue by applying the greedy agglomerative crossover operator with $r$ values chosen from the most favorable range.

In addition, the similarity in problem statements of the k-means, k-medoids and k-median problems promises us a reasonable hope for the applicability of the same approaches to improving the accuracy of algorithms, including VNS algorithms, by adjusting the parameter $r$ of the neighborhoods similar with GREEDY$_r$.

## 5. Conclusions

The process of introducing machine learning methods into all spheres of life determines the need to develop not only fast, but also the most accurate algorithms for solving related optimization problems. As practice shows, including this study, when solving some problems, the most popular clustering algorithm gives a result extremely far from the optimal k-means problem solution.

In this research, we introduced GREEDY$_r$ search neighborhoods and found that searching in both SWAP and GREEDY$_r$ neighborhoods has advantages over the simplest Lloyd's procedure. However, the results strongly depend on the parameters of such neighborhoods, and the optimal values of these parameters differ significantly for test problems. Nevertheless, searching in GREEDY$_r$ neighborhoods outperforms searching in SWAP neighborhoods in terms of accuracy.

We hope that our new variable neighborhood search algorithm (solver) for GPUs, which is more versatile due to its self-adjusting capability and has an advantage with respect to the accuracy of solving the k-means problem over known algorithms, will encourage researchers and practitioners in the field of machine learning to build competitive systems with the lowest possible error within a limited time. Such systems should be in demand when clustering geographic data, as well as when solving a wide range of problems with the highest cost of error.

**Author Contributions:** Conceptualization, L.K. and I.R.; methodology, L.K.; software, L.K.; validation, I.R. and E.T.; formal analysis, I.R. and A.P.; investigation, I.R.; resources, L.K. and E.T.; data curation, I.R.; writing—original draft preparation, L.K. and I.R.; writing—review and editing, L.K., E.T., and A.P.; visualization, I.R.; supervision, L.K.; project administration, L.K.; funding acquisition, L.K. and A.P. All authors have read and agreed to the published version of the manuscript.

**Funding:** This research was funded by The Ministry of Science and Higher Education of the Russian Federation, project No. FEFE-2020-0013.

**Conflicts of Interest:** The authors declare no conflict of interest. The funder had no role in the design of the study; in the collection, analyses, or interpretation of data; in the writing of the manuscript, or in the decision to publish the results.

## Abbreviations

The following abbreviations are used in this manuscript:

| | |
|---|---|
| NP | Non-deterministic polynomial-time |
| MSSC | Minimum Sum-of-Squares Clustering |
| SSE | Sum of Squared Errors |
| ALA algorithm | Alternate Location-Allocation algorithm |
| VNS | Variable Neighborhood Search |
| GA | Genetic Algorithm |
| IBC | Information Bottleneck Clustering |
| VND | Variable Neighborhood Descent |
| RVNS | Randomized Variable Neighborhood Search |
| GPU | Graphics Processing Unit |
| CPU | Central Processing Unit |
| RAM | Random Access Memory |
| CUDA | Compute Unified Device Architecture |
| IHEPC | Individual Household Electric Power Consumption |
| Lloyd-MS | Lloyd's procedure in a multi-start mode |
| J-means-MS | J-Means algorithm in a multi-start mode (SWAP$_1$+Lloyd VND) |
| GREEDY$_r$ | A neighborhood formed by applying greedy agglomerative procedures with $r$ excessive clusters, and the RVNS algorithm which combines search in such neighborhood with Lloyd's procedure |
| SWAP$_r$ | A neighborhood formed by replacing $r$ centroids by data vectors, and the RVNS algorithm which combines search in such neighborhood with Lloyd's procedure |
| GH-VNS1 | VNS algorithm with GREEDY$r$ neighborhoods and GREEDY$_1$ for the initial neighborhood type |
| GH-VNS2 | VNS algorithm with GREEDY$r$ neighborhoods and GREEDY$_{random}$ for the initial neighborhood type |
| GH-VNS3 | VNS algorithm with GREEDY$r$ neighborhoods and GREEDY$_k$ for the initial neighborhood type |
| GA-1 | Genetic algorithm with the single-point crossover, real-valued genes encoded by centroid positions, and the uniform random mutation |
| AdaptiveGreedy | New algorithm proposed in this article |

## Appendix A. Results of Computational Experiments

**Table A1.** Comparative results for Mopsi-Joensuu data set. 6014 data vectors in $\mathbb{R}^2$, $k = 30$ clusters, time limitation 5 s.

| Algorithm or Neighborhood | Achieved SSE Summarized After 30 Runs | | | | |
|---|---|---|---|---|---|
| | Min (Record) | Max (Worst) | Average | Median | Std.dev |
| Lloyd-MS | 35.5712 | 43.3993 | 39.1185 | 38.7718 | 2.9733 |
| j-Means-MS | 18.4076 | 23.7032 | 20.3399 | 19.8533 | 1.8603 |
| GREEDY$_1$ | 18.3253 | 27.6990 | 21.4555 | 21.6629 | 3.1291 |
| GREEDY$_2$ | 18.3253 | 21.7008 | 19.3776 | 18.3254 | 1.6119 |
| GREEDY$_3$ | **18.3145** | 21.7007 | 18.5817 | 18.3254 | 0.9372 |
| GREEDY$_5$ | 18.3253 | 21.7007 | 18.5129 | 18.3254 | 0.7956 |
| GREEDY$_7$ | 18.3253 | 21.7008 | 18.5665 | 18.3255 | 0.9021 |
| GREEDY$_{10}$ | 18.3253 | 21.7010 | 18.5666 | 18.3255 | 0.9021 |
| GREEDY$_{12}$ | 18.3254 | 21.7009 | 18.5852 | 18.3256 | 0.9362 |
| GREEDY$_{15}$ | 18.3254 | 18.3257 | 18.3255 | 18.3255 | **0.0001** |
| GREEDY$_{20}$ | 18.3254 | 18.3263 | 18.3257 | 18.3257 | 0.0002 |
| GREEDY$_{25}$ | 18.3254 | 18.3257 | 18.3255 | 18.3255 | **0.0001** |
| GREEDY$_{30}$ | 18.3254 | 18.3261 | 18.3258 | 18.3258 | 0.0002 |
| GH-VNS1 | 18.3147 | 18.3255 | **18.3238** | **18.3253** | 0.0039 |
| GH-VNS2 | 18.3253 | 21.7008 | 19.3776 | 18.3254 | 1.6119 |
| GH-VNS3 | 18.3146 | 21.6801 | 18.5634 | 18.3254 | 0.8971 |
| SWAP$_1$ (the best of SWAP$_r$) | 18.9082 | 20.3330 | 19.4087 | 18.9967 | 0.6019 |
| GA-1 | 18.6478 | 21.1531 | 19.9555 | 19.9877 | 0.6632 |
| AdaptiveGreedy | 18.3146 | 18.3258 | 18.3240 | **18.3253** | 0.0037 |

**Table A2.** Comparative results for Mopsi-Joensuu data set. 6014 data vectors in $\mathbb{R}^2$, $k = 100$ clusters, time limitation 5 s.

| Algorithm or Neighborhood | Achieved SSE Summarized After 30 Runs | | | | |
|---|---|---|---|---|---|
| | Min (Record) | Max (Worst) | Average | Median | Std.dev |
| Lloyd-MS | 23.1641 | 34.7834 | 27.5520 | 27.1383 | 3.6436 |
| j-Means-MS | **1.7628** | 31.8962 | 11.1832 | 2.4216 | 11.7961 |
| GREEDY$_1$ | 20.6701 | 35.5447 | 28.9970 | 29.2429 | 5.0432 |
| GREEDY$_2$ | 2.8264 | 29.0682 | 9.9708 | 5.3363 | 9.6186 |
| GREEDY$_3$ | 2.6690 | 10.5998 | 4.1444 | 3.0588 | 2.2108 |
| GREEDY$_5$ | 1.9611 | 4.3128 | 2.7385 | 2.7299 | 0.6135 |
| GREEDY$_7$ | 2.0837 | 4.6443 | 2.8730 | 2.6358 | 0.7431 |
| GREEDY$_{10}$ | 1.9778 | 3.8635 | 2.5613 | 2.3304 | 0.6126 |
| GREEDY$_{12}$ | 1.7817 | 4.3023 | 2.5639 | 2.2009 | 0.8730 |
| GREEDY$_{15}$ | 1.9564 | 3.1567 | 2.3884 | 2.2441 | 0.3620 |
| GREEDY$_{20}$ | 1.7937 | 3.2809 | 2.4542 | 2.3500 | 0.4746 |
| GREEDY$_{25}$ | 1.9532 | 3.3874 | 2.4195 | 2.2575 | 0.5470 |
| GREEDY$_{30}$ | 1.9274 | 2.4580 | 2.1723 | 2.1458 | 0.2171 |
| GREEDY$_{50}$ | 1.8903 | 9.3675 | 2.8047 | 2.1614 | 2.0838 |
| GREEDY$_{75}$ | 1.7878 | 2.8855 | 2.1775 | 2.0272 | 0.4023 |
| GREEDY$_{100}$ | 1.8021 | 2.2942 | 2.0158 | 1.9849 | 0.1860 |
| GH-VNS1 | 2.8763 | 17.1139 | 7.3196 | 4.3341 | 5.7333 |
| GH-VNS2 | 2.8264 | 29.0682 | 9.9708 | 5.3363 | 9.6186 |
| GH-VNS3 | 1.7643 | 2.7357 | 2.0513 | 1.9822 | 0.2699 |
| SWAP$_3$ (the best of rand. SWAP$_r$) | 4.9739 | 23.6572 | 9.0159 | 8.3907 | 4.1351 |
| GA-1 | 4.8922 | 19.1543 | 8.5914 | 7.1764 | 4.1096 |
| AdaptiveGreedy | 1.7759 | 2.3265 | **1.9578** | **1.9229** | **0.1523** |

**Table A3.** Comparative results for Mopsi-Joensuu data set. 6014 data vectors in $\mathbb{R}^2$, $k = 300$ clusters, time limitation 5 s.

| Algorithm or Neighborhood | Achieved SSE Summarized After 30 Runs | | | | |
|---|---|---|---|---|---|
| | Min (Record) | Max (Worst) | Average | Median | Std.dev |
| Lloyd-MS | 4.1789 | 14.7570 | 9.1143 | 9.3119 | 3.0822 |
| j-Means-MS | 7.0119 | 22.3126 | 14.2774 | 12.6199 | 5.5095 |
| GREEDY$_1$ | 7.1654 | 15.3500 | 9.6113 | 9.2176 | 2.5266 |
| GREEDY$_2$ | 4.9896 | 14.4839 | 8.9197 | 8.2013 | 3.3072 |
| GREEDY$_3$ | 5.8967 | 14.1110 | 8.3260 | 8.0441 | 2.2140 |
| GREEDY$_5$ | 2.9115 | 10.2536 | 5.8012 | 5.7305 | 2.2740 |
| GREEDY$_7$ | 2.6045 | 7.9868 | 4.4201 | 4.0548 | 1.4841 |
| GREEDY$_{10}$ | 2.5497 | 8.6758 | 4.1796 | 2.9639 | 1.8494 |
| GREEDY$_{12}$ | 2.0753 | 4.7134 | 3.0383 | 2.8777 | 0.8348 |
| GREEDY$_{15}$ | 1.8975 | 8.7890 | 3.8615 | 3.2661 | 1.8064 |
| GREEDY$_{20}$ | 1.1878 | 3.7944 | 2.4577 | 2.4882 | 0.9554 |
| GREEDY$_{25}$ | 1.1691 | 3.5299 | 1.8489 | 1.6407 | 0.7460 |
| GREEDY$_{30}$ | 1.1151 | 4.9425 | 2.3711 | 2.0582 | 1.1501 |
| GREEDY$_{50}$ | 1.3526 | 3.5471 | 1.8635 | 1.7114 | 0.6046 |
| GREEDY$_{75}$ | 1.0533 | 5.5915 | 1.9129 | 1.4261 | 1.2082 |
| GREEDY$_{100}$ | 0.8047 | 2.0349 | 1.2602 | 1.1994 | 0.3811 |
| GREEDY$_{150}$ | 0.6243 | 1.4755 | 0.8743 | 0.8301 | 0.2447 |
| GREEDY$_{200}$ | 0.4555 | 1.0154 | 0.6746 | 0.5882 | 0.2103 |
| GREEDY$_{250}$ | 0.4789 | 1.3368 | 0.7233 | 0.6695 | 0.2164 |
| GREEDY$_{300}$ | 0.5474 | 1.0472 | 0.7228 | 0.6657 | 0.1419 |
| GH-VNS1 | 1.6219 | 5.2528 | 3.0423 | 3.1332 | 1.0222 |
| GH-VNS2 | 1.2073 | 8.6144 | 3.2228 | 2.3501 | 2.4014 |
| GH-VNS3 | 0.4321 | 0.6838 | 0.6024 | 0.6139 | 0.0836 |
| SWAP$_{12}$ (the best of SWAP by median) | 2.6016 | 5.5038 | 3.6219 | 3.3612 | 1.0115 |
| SWAP$_{20}$ (the best of SWAP by avg.) | 2.1630 | 5.1235 | 3.4958 | 3.4076 | 0.8652 |
| GA-1 | 5.4911 | 12.6950 | 8.8799 | 7.7181 | 2.5384 |
| AdaptiveGreedy | **<u>0.3128</u>** | 0.6352 | **<u>0.4672</u>** | **<u>0.4604</u>** | **<u>0.1026</u>** |

**Table A4.** Comparative results for Mopsi-Finland data set. 13,467 data vectors in $\mathbb{R}^2$, $k = 30$ clusters, time limitation 5 s.

| Algorithm or Neighborhood | Achieved SSE Summarized After 30 Runs | | | | |
|---|---|---|---|---|---|
| | Min (Record) | Max (Worst) | Average | Median | Std.dev |
| Lloyd-MS | $4.79217 \times 10^{10}$ | $6.36078 \times 10^{10}$ | $5.74896 \times 10^{10}$ | $5.79836 \times 10^{10}$ | $3.69760 \times 10^9$ |
| j-Means-MS | $3.43535 \times 10^{10}$ | $4.26830 \times 10^{10}$ | $3.66069 \times 10^{10}$ | $3.60666 \times 10^{10}$ | $1.75725 \times 10^9$ |
| GREEDY$_1$ | $3.43195 \times 10^{10}$ | $3.70609 \times 10^{10}$ | $3.51052 \times 10^{10}$ | $3.48431 \times 10^{10}$ | $7.42\,636 \times 10^8$ |
| GREEDY$_2$ | $3.43194 \times 10^{10}$ | $3.49405 \times 10^{10}$ | $3.44496 \times 10^{10}$ | $3.44140 \times 10^{10}$ | $1.64\,360 \times 10^8$ |
| GREEDY$_3$ | $3.43195 \times 10^{10}$ | $3.49411 \times 10^{10}$ | $3.44474 \times 10^{10}$ | $3.44140 \times 10^{10}$ | $1.71131 \times 10^8$ |
| GREEDY$_5$ | $3.43195 \times 10^{10}$ | $3.48411 \times 10^{10}$ | $3.44663 \times 10^{10}$ | $3.44141 \times 10^{10}$ | $1.65153 \times 10^8$ |
| GREEDY$_7$ | $3.42531 \times 10^{10}$ | $3.47610 \times 10^{10}$ | $3.44091 \times 10^{10}$ | ***3.43504 × 10^{10}*** | $1.76023 \times 10^8$ |
| GREEDY$_{10}$ | $3.42560 \times 10^{10}$ | $3.48824 \times 10^{10}$ | $3.45106 \times 10^{10}$ | $3.43573 \times 10^{10}$ | $2.36526 \times 10^8$ |
| GREEDY$_{12}$ | $3.42606 \times 10^{10}$ | $3.48822 \times 10^{10}$ | $3.44507 \times 10^{10}$ | $3.43901 \times 10^{10}$ | $1.68986 \times 10^8$ |
| GREEDY$_{15}$ | $3.42931 \times 10^{10}$ | $3.47817 \times 10^{10}$ | ***3.43874 × 10^{10}*** | $3.43901 \times 10^{10}$ | $8.31510 \times 10^7$ |
| GREEDY$_{20}$ | $3.42954 \times 10^{10}$ | $3.48826 \times 10^{10}$ | $3.44186 \times 10^{10}$ | $3.43905 \times 10^{10}$ | $1.28972 \times 10^8$ |
| GREEDY$_{25}$ | $3.43877 \times 10^{10}$ | $3.44951 \times 10^{10}$ | $3.43982 \times 10^{10}$ | $3.43907 \times 10^{10}$ | **$2.57320 \times 10^7$** |
| GREEDY$_{30}$ | $3.43900 \times 10^{10}$ | $3.48967 \times 10^{10}$ | $3.45169 \times 10^{10}$ | $3.43979 \times 10^{10}$ | $1.93565 \times 10^8$ |
| GH-VNS1 | $3.42626 \times 10^{10}$ | $3.48724 \times 10^{10}$ | $3.45244 \times 10^{10}$ | $3.44144 \times 10^{10}$ | $2.00510 \times 10^8$ |
| GH-VNS2 | **$3.42528 \times 10^{10}$** | $3.48723 \times 10^{10}$ | $3.44086 \times 10^{10}$ | **$3.43474 \times 10^{10}$** | $1.54771 \times 10^8$ |
| GH-VNS3 | **<u>$3.42528 \times 10^{10}$</u>** | $3.47955 \times 10^{10}$ | **<u>$3.43826 \times 10^{10}$</u>** | **<u>$3.43474 \times 10^{10}$</u>** | $1.02356 \times 10^8$ |
| SWAP$_1$ (the best of SWAP$_r$) | $3.43199 \times 10^{10}$ | $3.55777 \times 10^{10}$ | $3.46821 \times 10^{10}$ | $3.46056 \times 10^{10}$ | $3.22711 \times 10^8$ |
| GA-1 | $3.48343 \times 10^{10}$ | $3.81846 \times 10^{10}$ | $3.65004 \times 10^{10}$ | $3.64415 \times 10^{10}$ | $1.00523 \times 10^9$ |
| AdaptiveGreedy | **<u>$3.42528 \times 10^{10}$</u>** | $3.47353 \times 10^{10}$ | **<u>$3.43385 \times 10^{10}$</u>** | **<u>$3.43473 \times 10^{10}$</u>** | **$1.03984 \times 10^8$** |

**Table A5.** Comparative results for Mopsi- Finland data set. 13,467 data vectors in $\mathbb{R}^2$, $k = 300$ clusters, time limitation 5 s.

| Algorithm or Neighborhood | Achieved SSE Summarized After 30 Runs | | | | |
|---|---|---|---|---|---|
| | Min (Record) | Max (Worst) | Average | Median | Std.dev |
| Lloyd-MS | $5.41643 \times 10^9$ | $6.89261 \times 10^9$ | $6.25619 \times 10^9$ | $6.24387 \times 10^9$ | $3.23827 \times 10^8$ |
| j-Means-MS | $6.75216 \times 10^8$ | $1.38889 \times 10^9$ | $8.92782 \times 10^8$ | $8.35397 \times 10^8$ | $1.86995 \times 10^8$ |
| GREEDY$_1$ | $4.08445 \times 10^9$ | $9.07208 \times 10^9$ | $5.89974 \times 10^9$ | $5.59903 \times 10^9$ | $1.47601 \times 10^8$ |
| GREEDY$_2$ | $1.11352 \times 10^9$ | $2.10247 \times 10^9$ | $1.59229 \times 10^9$ | $1.69165 \times 10^9$ | $2.89625 \times 10^8$ |
| GREEDY$_3$ | $9.63842 \times 10^8$ | $2.15674 \times 10^9$ | $1.61490 \times 10^9$ | $1.60123 \times 10^9$ | $3.06567 \times 10^8$ |
| GREEDY$_5$ | $9.11944 \times 10^8$ | $2.36799 \times 10^9$ | $1.66021 \times 10^9$ | $1.70448 \times 10^9$ | $3.68575 \times 10^8$ |
| GREEDY$_7$ | $1.17328 \times 10^9$ | $2.44476 \times 10^9$ | $1.77589 \times 10^9$ | $1.80948 \times 10^9$ | $2.68354 \times 10^8$ |
| GREEDY$_{10}$ | $1.14221 \times 10^9$ | $2.00426 \times 10^9$ | $1.67586 \times 10^9$ | $1.69601 \times 10^9$ | $2.14822 \times 10^8$ |
| GREEDY$_{12}$ | $9.41133 \times 10^8$ | $2.28940 \times 10^9$ | $1.59715 \times 10^9$ | $1.62288 \times 10^9$ | $3.01841 \times 10^8$ |
| GREEDY$_{15}$ | $8.86983 \times 10^8$ | $2.29776 \times 10^9$ | $1.53989 \times 10^9$ | $1.43319 \times 10^9$ | $3.70138 \times 10^8$ |
| GREEDY$_{20}$ | $1.02224 \times 10^9$ | $2.11636 \times 10^9$ | $1.62601 \times 10^9$ | $1.64029 \times 10^9$ | $2.45576 \times 10^8$ |
| GREEDY$_{25}$ | $9.07984 \times 10^8$ | $1.87134 \times 10^9$ | $1.42878 \times 10^9$ | $1.42864 \times 10^9$ | $2.74744 \times 10^8$ |
| GREEDY$_{30}$ | $8.44247 \times 10^8$ | $2.22882 \times 10^9$ | $1.50817 \times 10^9$ | $1.56015 \times 10^9$ | $3.52497 \times 10^8$ |
| GREEDY$_{50}$ | $7.98191 \times 10^8$ | $1.68198 \times 10^9$ | $1.26851 \times 10^9$ | $1.17794 \times 10^9$ | $2.67082 \times 10^8$ |
| GREEDY$_{75}$ | $6.97650 \times 10^8$ | $1.74139 \times 10^9$ | $1.16422 \times 10^9$ | $1.16616 \times 10^9$ | $2.82454 \times 10^8$ |
| GREEDY$_{100}$ | $6.55465 \times 10^8$ | $1.44162 \times 10^9$ | $1.03643 \times 10^9$ | $1.09001 \times 10^9$ | $1.95246 \times 10^8$ |
| GREEDY$_{150}$ | $5.94256 \times 10^8$ | $1.45317 \times 10^9$ | $8.88898 \times 10^8$ | $7.96787 \times 10^8$ | $2.33137 \times 10^8$ |
| GREEDY$_{200}$ | $5.60885 \times 10^8$ | $1.41411 \times 10^9$ | $7.96908 \times 10^8$ | $7.20282 \times 10^8$ | $2.26191 \times 10^8$ |
| GREEDY$_{250}$ | $5.58602 \times 10^8$ | $1.13946 \times 10^9$ | $7.58434 \times 10^8$ | $6.81196 \times 10^8$ | $1.65511 \times 10^8$ |
| GREEDY$_{300}$ | $5.68646 \times 10^8$ | $1.41338 \times 10^9$ | $7.35067 \times 10^8$ | $6.83004 \times 10^8$ | $1.76126 \times 10^8$ |
| GH-VNS1 | $1.40141 \times 10^9$ | $2.86919 \times 10^9$ | $2.16238 \times 10^9$ | $2.10817 \times 10^9$ | $3.42105 \times 10^8$ |
| GH-VNS2 | $8.22679 \times 10^8$ | $2.12228 \times 10^9$ | $1.40322 \times 10^9$ | $1.39457 \times 10^9$ | $2.96599 \times 10^8$ |
| GH-VNS3 | $\mathbf{5.33373 \times 10^8}$ | $7.29800 \times 10^8$ | $\mathbf{5.74914 \times 10^8}$ | $\mathbf{5.48427 \times 10^8}$ | $\mathbf{5.05346 \times 10^7}$ |
| SWAP$_1$ (the best of. SWAP$_r$) | $6.69501 \times 10^8$ | $9.06507 \times 10^8$ | $7.48932 \times 10^8$ | $7.35532 \times 10^8$ | $6.74846 \times 10^7$ |
| GA-1 | $4.54419 \times 10^9$ | $7.11460 \times 10^9$ | $5.67688 \times 10^9$ | $5.61135 \times 10^9$ | $5.99687 \times 10^8$ |
| AdaptiveGreedy | $\underline{\mathbf{5.27254 \times 10^8}}$ | $7.09410 \times 10^8$ | $\underline{\mathbf{5.60867 \times 10^8}}$ | $\underline{\mathbf{5.38952 \times 10^8}}$ | $\underline{4.89257 \times 10^7}$ |

**Table A6.** Comparative results for BIRCH3 data set. $10^5$ data vectors in $\mathbb{R}^2$, $k = 100$ clusters, time limitation 10 s.

| Algorithm or Neighborhood | Achieved SSE Summarized After 30 Runs | | | | |
|---|---|---|---|---|---|
| | Min (Record) | Max (Worst) | Average | Median | Std.dev |
| Lloyd-MS | $8.13022 \times 10^{13}$ | $9.51129 \times 10^{13}$ | $8.96327 \times 10^{13}$ | $9.06147 \times 10^{13}$ | $4.84194 \times 10^{12}$ |
| j-Means-MS | $4.14627 \times 10^{13}$ | $6.25398 \times 10^{13}$ | $4.78063 \times 10^{13}$ | $4.55711 \times 10^{13}$ | $6.89734 \times 10^{12}$ |
| GREEDY$_1$ | $3.73299 \times 10^{13}$ | $5.64559 \times 10^{13}$ | $4.13352 \times 10^{13}$ | $3.90845 \times 10^{13}$ | $5.19021 \times 10^{12}$ |
| GREEDY$_2$ | $3.71499 \times 10^{13}$ | $3.72063 \times 10^{13}$ | $3.71689 \times 10^{13}$ | $3.71565 \times 10^{13}$ | $2.44802 \times 10^{10}$ |
| GREEDY$_3$ | $3.71518 \times 10^{13}$ | $3.72643 \times 10^{13}$ | $3.71840 \times 10^{13}$ | $3.71545 \times 10^{13}$ | $4.12818 \times 10^{10}$ |
| GREEDY$_5$ | $3.71485 \times 10^{13}$ | $3.72087 \times 10^{13}$ | $\mathbf{3.71644 \times 10^{13}}$ | $\underline{\mathbf{3.71518 \times 10^{13}}}$ | $2.22600 \times 10^{10}$ |
| GREEDY$_7$ | $3.71518 \times 10^{13}$ | $3.72267 \times 10^{13}$ | $\underline{3.71755 \times 10^{13}}$ | $\underline{3.71658 \times 10^{13}}$ | $2.24845 \times 10^{10}$ |
| GREEDY$_{10}$ | $3.71555 \times 10^{13}$ | $3.72119 \times 10^{13}$ | $3.71771 \times 10^{13}$ | $3.71794 \times 10^{13}$ | $1.90289 \times 10^{10}$ |
| GREEDY$_{12}$ | $3.71556 \times 10^{13}$ | $3.72954 \times 10^{13}$ | $3.71892 \times 10^{13}$ | $3.71693 \times 10^{13}$ | $3.91673 \times 10^{10}$ |
| GREEDY$_{15}$ | $3.71626 \times 10^{13}$ | $3.72169 \times 10^{13}$ | $3.71931 \times 10^{13}$ | $3.71963 \times 10^{13}$ | $\mathbf{1.86102 \times 10^{10}}$ |
| GREEDY$_{20}$ | $3.71600 \times 10^{13}$ | $3.72638 \times 10^{13}$ | $3.72118 \times 10^{13}$ | $3.72153 \times 10^{13}$ | $\underline{2.69206 \times 10^{10}}$ |
| GREEDY$_{25}$ | $3.72042 \times 10^{13}$ | $3.72690 \times 10^{13}$ | $3.72284 \times 10^{13}$ | $3.72228 \times 10^{13}$ | $2.14437 \times 10^{10}$ |
| GREEDY$_{30}$ | $3.72180 \times 10^{13}$ | $3.73554 \times 10^{13}$ | $3.72586 \times 10^{13}$ | $3.72471 \times 10^{13}$ | $4.33818 \times 10^{10}$ |
| GREEDY$_{50}$ | $3.72166 \times 10^{13}$ | $3.76422 \times 10^{13}$ | $3.73883 \times 10^{13}$ | $3.73681 \times 10^{13}$ | $16.1061 \times 10^{10}$ |
| GREEDY$_{75}$ | $3.72399 \times 10^{13}$ | $3.84870 \times 10^{13}$ | $3.76286 \times 10^{13}$ | $3.74750 \times 10^{13}$ | $41.6632 \times 10^{10}$ |
| GREEDY$_{100}$ | $3.72530 \times 10^{13}$ | $3.91589 \times 10^{13}$ | $3.80730 \times 10^{13}$ | $3.84482 \times 10^{13}$ | $61.9706 \times 10^{10}$ |
| GH-VNS1 | $3.71914 \times 10^{13}$ | $3.77527 \times 10^{13}$ | $3.73186 \times 10^{13}$ | $3.72562 \times 10^{13}$ | $18.3590 \times 10^{10}$ |
| GH-VNS2 | $3.71568 \times 10^{13}$ | $3.73791 \times 10^{13}$ | $3.72116 \times 10^{13}$ | $3.72051 \times 10^{13}$ | $6.08081 \times 10^{10}$ |
| GH-VNS3 | $3.71619 \times 10^{13}$ | $3.73487 \times 10^{13}$ | $3.72387 \times 10^{13}$ | $3.72282 \times 10^{13}$ | $5.96618 \times 10^{10}$ |
| SWAP$_1$ (the best of SWAP$_r$) | $4.28705 \times 10^{13}$ | $5.48014 \times 10^{13}$ | $4.82383 \times 10^{13}$ | $4.75120 \times 10^{13}$ | $3.90128 \times 10^{12}$ |
| GA-1 | $3.84317 \times 10^{13}$ | $4.08357 \times 10^{13}$ | $3.97821 \times 10^{13}$ | $3.97088 \times 10^{13}$ | $7.43642 \times 10^{11}$ |
| AdaptiveGreedy | $\mathbf{3.71484 \times 10^{13}}$ | $3.72011 \times 10^{13}$ | $3.71726 \times 10^{13}$ | $3.71749 \times 10^{13}$ | $2.02784 \times 10^{10}$ |

**Table A7.** Comparative results for BIRCH3 data set. $10^5$ data vectors in $\mathbb{R}^2$, $k = 300$ clusters, time limitation 10 s.

| Algorithm or Neighborhood | Achieved SSE Summarized After 30 Runs | | | | |
|---|---|---|---|---|---|
| | **Min (Record)** | **Max (Worst)** | **Average** | **Median** | **Std.dev** |
| Lloyd-MS | $3.49605 \times 10^{13}$ | $4.10899 \times 10^{13}$ | $3.74773 \times 10^{13}$ | $3.77191 \times 10^{13}$ | $2.32012 \times 10^{12}$ |
| j-Means-MS | $1.58234 \times 10^{13}$ | $2.02926 \times 10^{13}$ | $1.75530 \times 10^{13}$ | $1.70507 \times 10^{13}$ | $1.43885 \times 10^{12}$ |
| GREEDY$_1$ | $1.48735 \times 10^{13}$ | $2.63695 \times 10^{13}$ | $1.71372 \times 10^{13}$ | $1.60354 \times 10^{13}$ | $2.98555 \times 10^{12}$ |
| GREEDY$_2$ | $1.31247 \times 10^{13}$ | $1.45481 \times 10^{13}$ | $1.37228 \times 10^{13}$ | $1.36745 \times 10^{13}$ | $4.01697 \times 10^{11}$ |
| GREEDY$_3$ | $1.34995 \times 10^{13}$ | $1.49226 \times 10^{13}$ | $1.39925 \times 10^{13}$ | $1.39752 \times 10^{13}$ | $4.85917 \times 10^{11}$ |
| GREEDY$_5$ | $1.33072 \times 10^{13}$ | $1.45757 \times 10^{13}$ | $1.39069 \times 10^{13}$ | $1.38264 \times 10^{13}$ | $4.46890 \times 10^{11}$ |
| GREEDY$_7$ | $1.34959 \times 10^{13}$ | $1.49669 \times 10^{13}$ | $1.41606 \times 10^{13}$ | $1.41764 \times 10^{13}$ | $4.92200 \times 10^{11}$ |
| GREEDY$_{10}$ | $1.31295 \times 10^{13}$ | $1.42722 \times 10^{13}$ | $1.35970 \times 10^{13}$ | $1.35318 \times 10^{13}$ | $3.70511 \times 10^{11}$ |
| GREEDY$_{12}$ | $1.32677 \times 10^{13}$ | $1.49028 \times 10^{13}$ | $1.35561 \times 10^{13}$ | $1.33940 \times 10^{13}$ | $4.44283 \times 10^{11}$ |
| GREEDY$_{15}$ | $1.32077 \times 10^{13}$ | $1.41079 \times 10^{13}$ | $1.34102 \times 10^{13}$ | $1.33832 \times 10^{13}$ | $2.16247 \times 10^{11}$ |
| GREEDY$_{20}$ | $1.31994 \times 10^{13}$ | $1.43160 \times 10^{13}$ | $1.35420 \times 10^{13}$ | $1.34096 \times 10^{13}$ | $3.43684 \times 10^{11}$ |
| GREEDY$_{25}$ | $1.31078 \times 10^{13}$ | $1.37699 \times 10^{13}$ | $1.33571 \times 10^{13}$ | $1.33040 \times 10^{13}$ | $2.16378 \times 10^{11}$ |
| GREEDY$_{30}$ | $1.32947 \times 10^{13}$ | $1.45967 \times 10^{13}$ | $1.37618 \times 10^{13}$ | $1.36729 \times 10^{13}$ | $3.92767 \times 10^{11}$ |
| GREEDY$_{50}$ | $1.32284 \times 10^{13}$ | $1.38691 \times 10^{13}$ | $1.34840 \times 10^{13}$ | $1.33345 \times 10^{13}$ | $2.70770 \times 10^{11}$ |
| GREEDY$_{75}$ | $1.30808 \times 10^{13}$ | $1.33266 \times 10^{13}$ | $1.31857 \times 10^{13}$ | $1.31833 \times 10^{13}$ | $7.22941 \times 10^{10}$ |
| GREEDY$_{100}$ | $1.30852 \times 10^{13}$ | $1.32697 \times 10^{13}$ | $1.31250 \times 10^{13}$ | $1.31067 \times 10^{13}$ | $4.94315 \times 10^{10}$ |
| GREEDY$_{150}$ | $1.30754 \times 10^{13}$ | $1.31446 \times 10^{13}$ | $1.30971 \times 10^{13}$ | $1.30952 \times 10^{13}$ | $1.82873 \times 10^{10}$ |
| GREEDY$_{200}$ | $1.30773 \times 10^{13}$ | $1.31172 \times 10^{13}$ | $\mathbf{1.30916 \times 10^{13}}$ | $\mathbf{1.30912 \times 10^{13}}$ | $1.08001 \times 10^{10}$ |
| GREEDY$_{250}$ | $1.30699 \times 10^{13}$ | $1.31073 \times 10^{13}$ | $\underline{1.30944 \times 10^{13}}$ | $\underline{1.30990 \times 10^{13}}$ | $1.18367 \times 10^{10}$ |
| GREEDY$_{300}$ | $\mathbf{1.30684 \times 10^{13}}$ | $1.31068 \times 10^{13}$ | $1.30917 \times 10^{13}$ | $1.30933 \times 10^{13}$ | $1.21748 \times 10^{10}$ |
| GH-VNS1 | $1.40452 \times 10^{13}$ | $1.56256 \times 10^{13}$ | $1.45212 \times 10^{13}$ | $1.42545 \times 10^{13}$ | $55.7231 \times 10^{10}$ |
| GH-VNS2 | $1.32287 \times 10^{13}$ | $1.38727 \times 10^{13}$ | $1,34654 \times 10^{13}$ | $1,34568 \times 10^{13}$ | $2,01065 \times 10^{11}$ |
| GH-VNS3 | $1.30996 \times 10^{13}$ | $1.31378 \times 10^{13}$ | $1.31158 \times 10^{13}$ | $1.31138 \times 10^{13}$ | $1.44998 \times 10^{10}$ |
| SWAP$_2$ (the best of SWAP$_r$ by median) | $2.18532 \times 10^{13}$ | $3.25705 \times 10^{13}$ | $2.54268 \times 10^{13}$ | $2.37312 \times 10^{13}$ | $3.78491 \times 10^{12}$ |
| SWAP$_7$ (the best of SWAP$_r$ by avg.) | $2.24957 \times 10^{13}$ | $2.86883 \times 10^{13}$ | $2.46775 \times 10^{13}$ | $2.47301 \times 10^{13}$ | $1.51198 \times 10^{12}$ |
| GA-1 | $1.38160 \times 10^{13}$ | $1.71472 \times 10^{13}$ | $1.55644 \times 10^{13}$ | $1.54336 \times 10^{13}$ | $9.21217 \times 10^{11}$ |
| AdaptiveGreedy | $1.30807 \times 10^{13}$ | $1.31113 \times 10^{13}$ | $1.30922 \times 10^{13}$ | $1.30925 \times 10^{13}$ | $\underline{\mathbf{0.87731 \times 10^{10}}}$ |

**Table A8.** Comparative results for S1 data set. 5000 data vectors in $\mathbb{R}^2$, $k = 15$ clusters, time limitation 1 s.

| Algorithm or Neighborhood | Achieved SSE Summarized After 30 Runs | | | | |
|---|---|---|---|---|---|
| | **Min (Record)** | **Max (Worst)** | **Average** | **Median** | **Std.dev** |
| Lloyd-MS | $\mathbf{8.91703 \times 10^{12}}$ | $8.91707 \times 10^{12}$ | $8.91704 \times 10^{12}$ | $8.91703 \times 10^{12}$ | $1.31098 \times 10^{7}$ |
| j-Means-MS | $\underline{\mathbf{8.91703 \times 10^{12}}}$ | $14.2907 \times 10^{12}$ | $12.1154 \times 10^{12}$ | $13.3667 \times 10^{12}$ | $2.38947 \times 10^{12}$ |
| GREEDY$_1$ | $\mathbf{8.91703 \times 10^{12}}$ | $13.2502 \times 10^{12}$ | $9.27814 \times 10^{12}$ | $\mathbf{8.91703 \times 10^{12}}$ | $1.25086 \times 10^{12}$ |
| GREEDY$_2$ | $\mathbf{8.91703 \times 10^{12}}$ | $8.91703 \times 10^{12}$ | $\mathbf{8.91703 \times 10^{12}}$ | $\mathbf{8.91703 \times 10^{12}}$ | $\mathbf{0.00000}$ |
| GREEDY$_3$ | $\mathbf{8.91703 \times 10^{12}}$ | $8.91703 \times 10^{12}$ | $\underline{\mathbf{8.91703 \times 10^{12}}}$ | $\mathbf{8.91703 \times 10^{12}}$ | $\underline{\mathbf{0.00000}}$ |
| GREEDY$_5$ | $\mathbf{8.91703 \times 10^{12}}$ | $8.91703 \times 10^{12}$ | $\mathbf{8.91703 \times 10^{12}}$ | $\mathbf{8.91703 \times 10^{12}}$ | $4.03023 \times 10^{5}$ |
| GREEDY$_7$ | $\mathbf{8.91703 \times 10^{12}}$ | $8.91703 \times 10^{12}$ | $\underline{\mathbf{8.91703 \times 10^{12}}}$ | $\mathbf{8.91703 \times 10^{12}}$ | $4.87232 \times 10^{5}$ |
| GREEDY$_{10}$ | $\mathbf{8.91703 \times 10^{12}}$ | $8.91703 \times 10^{12}$ | $\mathbf{8.91703 \times 10^{12}}$ | $\mathbf{8.91703 \times 10^{12}}$ | $5.12234 \times 10^{5}$ |
| GREEDY$_{12}$ | $\mathbf{8.91703 \times 10^{12}}$ | $8.91703 \times 10^{12}$ | $\underline{\mathbf{8.91703 \times 10^{12}}}$ | $\mathbf{8.91703 \times 10^{12}}$ | $3.16158 \times 10^{5}$ |
| GREEDY$_{15}$ | $\mathbf{8.91703 \times 10^{12}}$ | $8.91703 \times 10^{12}$ | $\underline{\mathbf{8.91703 \times 10^{12}}}$ | $\mathbf{8.91703 \times 10^{12}}$ | $5.01968 \times 10^{5}$ |
| GH-VNS1 | $\mathbf{8.91703 \times 10^{12}}$ | $8.91703 \times 10^{12}$ | $\mathbf{8.91703 \times 10^{12}}$ | $\mathbf{8.91703 \times 10^{12}}$ | $\mathbf{0.00000}$ |
| GH-VNS2 | $\mathbf{8.91703 \times 10^{12}}$ | $8,91703 \times 10^{12}$ | $\mathbf{8,91703 \times 10^{12}}$ | $\mathbf{8.91703 \times 10^{12}}$ | $\underline{\mathbf{0.00000}}$ |
| GH-VNS3 | $\mathbf{8.91703 \times 10^{12}}$ | $8.91703 \times 10^{12}$ | $\underline{\mathbf{8.91703 \times 10^{12}}}$ | $\mathbf{8.91703 \times 10^{12}}$ | $4.03023 \times 10^{5}$ |
| SWAP$_1$ (the best of SWAP) | $\mathbf{8.91703 \times 10^{12}}$ | $8.91709 \times 10^{12}$ | $8.91704 \times 10^{12}$ | $\mathbf{8.91703 \times 10^{12}}$ | $8.67594 \times 10^{6}$ |
| GA-1 | $\mathbf{8.91703 \times 10^{12}}$ | $8.91707 \times 10^{12}$ | $\mathbf{8.91703 \times 10^{12}}$ | $\mathbf{8.91703 \times 10^{12}}$ | $9.04519 \times 10^{6}$ |
| AdaptiveGreedy | $\mathbf{8.91703 \times 10^{12}}$ | $8.91703 \times 10^{12}$ | $\underline{\mathbf{8.91703 \times 10^{12}}}$ | $\underline{\mathbf{8.91703 \times 10^{12}}}$ | $\mathbf{0.00000}$ |

**Table A9.** Comparative results for S1 data set. 5000 data vectors in $\mathbb{R}^2$, $k = 50$ clusters, time limitation 1 s.

| Algorithm or Neighborhood | Achieved SSE Summarized After 30 Runs | | | | |
|---|---|---|---|---|---|
| | **Min (Record)** | **Max (Worst)** | **Average** | **Median** | **Std.dev** |
| Lloyd-MS | $3.94212 \times 10^{12}$ | $4.06133 \times 10^{12}$ | $3.99806 \times 10^{12}$ | $3.99730 \times 10^{12}$ | $4.52976 \times 10^{10}$ |
| j-Means-MS | $3.96626 \times 10^{12}$ | $4.40078 \times 10^{12}$ | $4.12311 \times 10^{12}$ | $4.07123 \times 10^{12}$ | $14.81090 \times 10^{10}$ |
| GREEDY$_1$ | $\mathit{3.82369 \times 10^{12}}$ | $4.19102 \times 10^{12}$ | $3.91601 \times 10^{12}$ | $3.88108 \times 10^{12}$ | $9.82433 \times 10^{10}$ |
| GREEDY$_2$ | $\mathit{3.74350 \times 10^{12}}$ | $3.76202 \times 10^{12}$ | $3.75014 \times 10^{12}$ | $3.74936 \times 10^{12}$ | $6.10139 \times 10^{9}$ |
| GREEDY$_3$ | $\mathit{3.74776 \times 10^{12}}$ | $3.76237 \times 10^{12}$ | $3.75455 \times 10^{12}$ | $3.75456 \times 10^{12}$ | $5.24513 \times 10^{9}$ |
| GREEDY$_5$ | $\mathit{3.74390 \times 10^{12}}$ | $3.77031 \times 10^{12}$ | $3.75345 \times 10^{12}$ | $3.75298 \times 10^{12}$ | $7.17733 \times 10^{9}$ |
| GREEDY$_7$ | $\mathit{3.74446 \times 10^{12}}$ | $3.77208 \times 10^{12}$ | $3.75277 \times 10^{12}$ | $3.75190 \times 10^{12}$ | $7.40052 \times 10^{9}$ |
| GREEDY$_{10}$ | $\mathit{3.74493 \times 10^{12}}$ | $3.76031 \times 10^{12}$ | $3.75159 \times 10^{12}$ | $3.75185 \times 10^{12}$ | $5.26553 \times 10^{9}$ |
| GREEDY$_{15}$ | $\mathit{3.74472 \times 10^{12}}$ | $3.77922 \times 10^{12}$ | $3.75426 \times 10^{12}$ | $3.75519 \times 10^{12}$ | $9.79855 \times 10^{9}$ |
| GREEDY$_{20}$ | $\mathit{3.75028 \times 10^{12}}$ | $3.76448 \times 10^{12}$ | $3.75586 \times 10^{12}$ | $3.75573 \times 10^{12}$ | $\mathbf{3.97310 \times 10^{9}}$ |
| GREEDY$_{25}$ | $\mathit{3.74770 \times 10^{12}}$ | $3.76224 \times 10^{12}$ | $3.75500 \times 10^{12}$ | $3.75572 \times 10^{12}$ | $4.95370 \times 10^{9}$ |
| GREEDY$_{30}$ | $3.75014 \times 10^{12}$ | $3.76010 \times 10^{12}$ | $3.75583 \times 10^{12}$ | $3.75661 \times 10^{12}$ | $3.45280 \times 10^{9}$ |
| GREEDY$_{50}$ | $3.74676 \times 10^{12}$ | $3.77396 \times 10^{12}$ | $3.76021 \times 10^{12}$ | $3.75933 \times 10^{12}$ | $9.09159 \times 10^{9}$ |
| GH-VNS1 | $\mathbf{3.74310 \times 10^{12}}$ | $3.76674 \times 10^{12}$ | $3.74911 \times 10^{12}$ | $\mathbf{3.74580 \times 10^{12}}$ | $6.99859 \times 10^{9}$ |
| GH-VNS2 | $3{,}75106 \times 10^{12}$ | $3{,}77369 \times 10^{12}$ | $3{,}75792 \times 10^{12}$ | $3{,}75782 \times 10^{12}$ | $6{,}67960 \times 10^{9}$ |
| GH-VNS3 | $3.75923 \times 10^{12}$ | $3.77964 \times 10^{12}$ | $3.76722 \times 10^{12}$ | $3.76812 \times 10^{12}$ | $6.00125 \times 10^{9}$ |
| SWAP$_3$ (the best of SWAP) | $3.75128 \times 10^{12}$ | $3.79170 \times 10^{12}$ | $3.77853 \times 10^{12}$ | $3.77214 \times 10^{12}$ | $4.53608 \times 10^{9}$ |
| GA-1 | $3.84979 \times 10^{12}$ | $3.99291 \times 10^{12}$ | $3.92266 \times 10^{12}$ | $3.92818 \times 10^{12}$ | $4.56845 \times 10^{12}$ |
| AdaptiveGreedy | $3.74340 \times 10^{12}$ | $3.76313 \times 10^{12}$ | $\underline{\mathbf{3.74851 \times 10^{12}}}$ | $3.75037 \times 10^{12}$ | $5.56298 \times 10^{9}$ |

**Table A10.** Comparative results for Individual Household Electric Power Consumption (IHEPC) data set. 2,075,259 data vectors in $\mathbb{R}^7$, $k = 15$ clusters, time limitation 5 min.

| Algorithm or Neighborhood | Achieved SSE Summarized After 30 Runs | | | | |
|---|---|---|---|---|---|
| | **Min (Record)** | **Max (Worst)** | **AVERAGE** | **Median** | **Std.dev** |
| Lloyd-MS | 12,874.8652 | 12,880.0703 | 12,876.0219 | 12,874.8652 | 2.2952 |
| j-Means-MS | 12,874.8652 | 13,118.6455 | 12,984.7081 | 12,962.1323 | 75.6539 |
| all GREEDY$_{1-15}$ (equal results) | **12,874.8633** | 12,874.8633 | **12,874.8633** | **12,874.8633** | **0.0000** |
| GH-VNS1 | **12,874.8633** | 12,874.8633 | **12,874.8633** | **12,874.8633** | **0.0000** |
| GH-VNS2 | **12,874.8633** | 12,874.8633 | **12,874.8633** | **12,874.8633** | **0.0000** |
| GH-VNS3 | **12,874.8633** | 12,874.8633 | **12,874.8633** | **12,874.8633** | **0.0000** |
| GA-1 | 12,874.8643 | 12,874.8652 | 12,874.8644 | 12,874.8643 | 0.0004 |
| AdaptiveGreedy | **12,874.8633** | 12,874.8633 | **12,874.8633** | **12,874.8633** | **0.0000** |

**Table A11.** Comparative results for Individual Household Electric Power Consumption (IHEPC) data set. 2,075,259 data vectors in $\mathbb{R}^7$, $k = 50$ clusters, time limitation 5 min.

| Algorithm or Neighborhood | Achieved SSE Summarized After 30 Runs | | | | |
|---|---|---|---|---|---|
| | **Min (Record)** | **Max (Worst)** | **Average** | **Median** | **Std.dev** |
| Lloyd-MS | 5605.0625 | 5751.1982 | 5671.0820 | 5660.4429 | 54.2467 |
| j-Means-MS | 5160.2700 | 6280.6440 | 5496.6539 | 5203.5679 | 493.7311 |
| GREEDY$_1$ | 5200.9268 | 5431.3647 | 5287.4101 | 5281.7300 | 77.0460 |
| GREEDY$_2$ | 5167.1482 | 5283.3894 | 5171.6509 | 5192.1274 | 7.7203 |
| GREEDY$_3$ | 5155.5166 | 5178.4063 | 5166.5360 | 5164.6045 | 8.1580 |
| GREEDY$_5$ | 5164.6040 | 5178.4336 | 5170.8829 | 5174.0938 | 6.0904 |
| GREEDY$_7$ | 5162.5381 | 5178.1269 | 5168.7218 | 5171.8292 | 6.4518 |
| GREEDY$_{10}$ | 5154.2017 | 5176.4502 | 5162.0460 | 5160.4014 | 7.2029 |
| GREEDY$_{12}$ | 5162.8715 | 5181.0281 | 5166.8952 | 5165.3295 | 6.0172 |
| GREEDY$_{15}$ | 5163.2500 | 5181.1333 | 5167.3385 | 5165.8037 | 5.7910 |
| GREEDY$_{20}$ | 5156.2852 | 5176.6855 | 5166.2013 | 5164.6323 | 7.8749 |

**Table A11.** *Cont.*

| Algorithm or Neighborhood | Achieved SSE Summarized After 30 Runs | | | | |
| --- | --- | --- | --- | --- | --- |
| | Min (Record) | Max (Worst) | Average | Median | Std.dev |
| GREEDY$_{25}$ | 5166.9820 | 5181.8529 | 5175.0317 | 5176.2136 | 6.1471 |
| GREEDY$_{30}$ | 5168.6309 | 5182.4351 | 5175.2414 | 5176.4512 | 6.4635 |
| GREEDY$_{50}$ | 5168.3887 | 5182.4321 | 5177.5249 | 5177.6855 | 5.4437 |
| GH-VNS1 | 5155.5166 | 5164.6313 | 5158.6549 | 5157.6812 | 3.7467 |
| GH-VNS2 | 5159.8818 | 5176.6855 | 5167.3365 | 5166.9512 | 5.6808 |
| GH-VNS3 | 5171.2969 | 5182.4321 | 5175.0468 | 5174.0752 | 3.6942 |
| GA-1 | 5215.9521 | 5248.4521 | 5230.2839 | 5226.0386 | 13.2694 |
| AdaptiveGreedy | **5153.5640** | 5163.9316 | **5157.0822** | **5155.5198** | **3.6034** |

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
