# Peer review of "Self-Adjusting Variable Neighborhood Search Algorithm for Near-Optimal k-Means Clustering"

_computation, doi:10.3390/computation8040090_

Round 1

Reviewer 1 Report

 The proposal is appealing and interesting, and the method deserves some consideration. Moreover, the paper is almost well written and well organized.

- The text, in general, reads well, but a grammatical revision could improve it further.

- The paper adequately puts the progress it reports in the context of previous work, representative referencing and first discussion.

- The authors could highlight better the new scientific contribution, for instance, analyzing several recent literature works.

Author Response

Dear reviewer,

Thank you very much for your review.

Concerning your comments:

  • The text, in general, reads well, but a grammatical revision could improve it further.

We applied to Ruth Miskell, as a native English speaker born in UK, for assistance. We hope that the grammar in this version is better.

  • The authors could highlight better the new scientific contribution, for instance, analyzing several recent literature works.

We reviewed the whole bibliography, replaced some very old and hardly relevant references, and added several new ones. Examples of new references are are:

[31A] Cohen-Addad, V.;Klein, P.N.;Mathieu, C. Local search yields approximation schemes for k-means and k-median in Euclidean and minor-free metrics. IEEE 2016 IEEE 57th Annual Symposium on Foundations of Computer Science (FOCS).2016, pp.353-364.DOI:10.1109/focs.2016.46

[32A] Hromkovic, J. Algorithmics for Hard Problems: Introduction to Combinatorial Optimization, Randomization, Approximation, and Heuristics. Springer, 2011.

[33A] Ng, T. Expanding Neighborhood Tabu Search for facility location problems in water infrastructure planning.Proceedings of the 2014 IEEE International Conference on Systems, Man, and Cybernetics (SMC), 2014, pp.3851-3854. DOI: 10.1109/smc.2014.6974531

[53A]Hansen, P.; Mladenovic, N. Variable Neighborhood Search. In: Martí R., Pardalos P., Resende M. (eds) Handbook of Heuristics. Springer, Cham., 2018  https://doi.org/10.1007/978-3-319-07124-4_19

[Nikolaev] Nikolaev, A.; Mladenovic, N.; Todosijevic, R. J-means and I-means for minimum sum-of-squares clustering on networks.Optimization Letters 2017, Vol.11(2), pp.359-376DOI: 10.1007/s11590-015-0974-4

[Frackiewicz] Frackiewicz , M.; Mandrella, A.; Palus, H. Fast Color Quantization by K-Means Clustering Combined with Image Sampling. Symmetry 2019, Vol. 11(8), Article ID 963. DOI: 10.3390/sym11080963

[Zhang] Zhang, G.; Li, Y.; Deng, X. K-Means Clustering-Based Electrical Equipment Identification for SmartBuilding Application.Information 2020, Vol.11(1),Article ID  27. DOI: 10.3390/info11010027.

[Qin] Qin, J.; Fu, W.; Gao, H.; Zheng, W.X. Distributed k -means algorithm and fuzzy c -means algorithm for sensor networks based on multiagent consensus theory. IEEE Trans. Cybern. 2016, Vol.47, pp. 772–783.

[Duarte] Duarte A., Mladenović N., Sánchez-Oro J., Todosijević R. Variable Neighborhood Descent. In: Martí R., Panos P., Resende M. (eds) Handbook of Heuristics. Springer, Cham. 2016. DOI:https://doi.org/10.1007/978-3-319-07153-4_9-1

[Xu] Xu, T.S.; Chiang, H.D.; Liu, G.Y.; Tan, C.W. Hierarchical k-means method for clustering large-scale advanced metering infrastructure data.IEEE Trans. Power Deliv.2015, Vol.32, pp.609–616

[Liu] Liu, H.; Wu, J.; Liu, T.; Tao, D.; Fu, Y. Spectral ensemble clustering via weighted k-means: Theoretical andpractical evidence. IEEE Trans. Knowl. Data Eng. 2017, Vol.29, pp.1129–1143.

[Yang] Yang, J.; Wang, J. Tag clustering algorithm lmmsk: Improved k-means algorithm based on latentsemantic analysis. J. Syst. Electron. 2017, Vol.28, pp.374–384.

[Wang] Wang, X.D.; Chen, R.C.; Yan, F.; Zeng, Z.Q.; Hong, C.Q. Fast adaptive k-means subspace clustering for high-dimensional data.IEEE Access2019, Vol.7, pp.639–651.

[Gu] Gu, Y.; Li, K.; Guo, Z.; Wang, Y. Semi-supervised k-means ddos detection method using hybrid featureselection algorithm. IEEE Access2019, Vol.7, pp.351–365.

[Chen] Chen, F.; Yang, Y.; Xu, L.; Zhang, T.; Zhang, Y. Big-data clustering: K-means or K-indicators? 2019, Preprint. Available online: https://arxiv.org/pdf/1906.00938.pdf (Accessed October 18, 2020)

[Hedar] Hedar, A.-R.; Ibrahim, A.-M.M.; Abdel-Hakim, A.E.; Sewisy, A.A. K-Means Cloning: Adaptive Spherical K-Means Clustering. Algorithms 2018, Vol.11, p.151 DOI:  https://doi.org/10.3390/a11100151

[Singh] Singh,N.; Singh, D.P.; Pant, B. ACOCA: Ant Colony Optimization Based Clustering Algorithm for Big Data Preprocessing. Int. J. Math. Eng. Manag.Sci.2019, Vol. 4(5), pp.1239–1250. DOI: https://dx.doi.org/10.33889/IJMEMS.2019.4.5-098

[89A] Miskovic, S.; Stanimirovich, Z.; Grujicic, I.An efficient variable neighborhood search for solving a robust dynamic facility location problem in emergency service network. Electronic Notes in Discrete Mathematics, 2015, Vol.47, pp.261-268.

[91A] Kratica, J.; Leitner, M.; Ljubic, I. Variable Neighborhood Search for Solving the Balanced Location Problem. Electronic Notes in Discrete Mathematics, 2012, Vol.39, pp.21-28. DOI: 10.1016/j.endm.2012.10.004

[95A] Wen, M.; Krapper, E.; Larsen, J.; Stidsen, T.K.A multilevel variable neighborhood search heuristic for a practical vehicle routing and driver scheduling problem.Networks2011, Vol.58(4), pp.311-323.

[Guo] Guo, X.; Zhang, X.; He, Y.; Jin, Y.; Qin, H.; Azhar, M.; Huang, J. Z. A Robust k-Means Clustering Algorithm Based on Observation Point Mechanism. Complexity 2020, Vol.2020, Article ID 3650926.https://doi.org/10.1155/2020/3650926

[Pham] Pham, D. T.; Afify, A. A. Clustering techniques and their applications in engineering. Proceedings of the Institution of Mechanical Engineers, Part C. Journal of Mechanical Engineering Science 2007, Vol. 221(11), pp. 1445–1459. DOI:10.1243/09544062jmes508.

[New8] Jain, A. K.; Murty, M. N.; Flynn, P. J. Data clustering: a review. ACM Comput.Surv. 1999, Vol. 31(3), pp. 264–323. DOI: 10.1145/331499.331504.

[New131] Afify, A. A.; Dimov, S.; Naim, M. M.; Valeva,V. Detecting cyclical disturbances in supply networks using data mining techniques. In Proceedings of the 2nd European Conference onManagement of Technology, Birmingham, UK, 2006, pp. 1–8.DOI: 10.1243/09544054JEM879.

[Naranjo] Naranjo, J. E.; Saha, R.; Tariq, M. T.; Hadi, M.; Xiao, Y.Pattern Recognition Using Clustering Analysis to Support Transportation System Management, Operations, and Modeling.Journal of Advanced Transportation. 2019, 2019, Article ID 1628417. DOI: 10.1155/2019/1628417.

[Kadir] Kadir, R. A.; Shima, Y.; Sulaiman, R.;  Ali, F. Clustering of public transport operation using K-means, 2018 IEEE 3rd International Conference on Big Data Analysis (ICBDA), 2018, pp. 427-532.

[Sesham] Sesham, A.; Padmanabham, P.; Govardhan, A. Application of Factor Analysis to k-means Clustering Algorithm on Transportation Data. International Journal of Computer Applications 2014, Vol. 95(15), pp.40-46, DOI: 10.5120/16673-6677.

[Nath ]Deb Nath, R. P.; Lee, H.-J.; Chowdhury, N. K.; Chang, J.-W. Modified K-Means Clustering for Travel Time Prediction Based on Historical Traffic Data. LNCS 2010, Vol. 6276, pp. 511–521. doi:10.1007/978-3-642-15387-7_55

[Montazeri] Montazeri-Gh, M.; A. Fotouhi, A. Traffic condition recognition using the k-means clustering method, Scientia Iranica 2011, Vol. 18(4), pp. 930-937. DOI:10.1016/j.scient.2011.07.004.

In addition, in accordance with the comments of the reviewers as well as in accordance with out own analysis, we:

  • added subsections in Sections 1 and 2;
  • added one new "geographical" dataset to highlight the efficiency of the new algorithm on such datasets;
  • added one new (genetic) algorithm for comparison;
  • added omited results of searching in SWAPr neighborhoods;
  • added some details of program realization;
  • tried to improve the description of our motivation and impact.

We hope that the present version of the paper is better.

Reviewer 2 Report

All is as in the report

Author Response

Dear Reviewer,

Thank you very much for your review.

Concerning your comment:

Maybe add in the list of references the following paper: B. Rosi ́c, S. Raden-ovic, L.J. Jankovic, M. Milojevic,Optimisation of planetary gear train usingmultiobjective genetic algorithm, Journal of the Balkan Tribological Association, Vol. 17, No 3, 462-475 (2011).

Dear Reviewer, we carefully read the abovementioned paper. However, in this research, we did not implement the multicriteria Pareto optimization nor genetic algorithms  which were implemented in the above mentioned paper. We focused on a single criterion which is the objective function. We provide experiments with genetic algorithms for the location problems including multicriteria approach, and the mentioned paper will be interesting for our current and further research in these fields.

In addition, in accordance with the comments of the reviewers as well as in accordance with out own analysis, we:

  • added subsections in Sections 1 and 2;
  • added one new "geographical" dataset to highlight the efficiency of the new algorithm on such datasets;
  • added one new (genetic) algorithm for comparison;
  • added omited results of searching in SWAPr neighborhoods;
  • added some details of program realization;
  • tried to improve the description of our motivation and impact;
  • reviewed the bibliography, replaced some old and hardly relevant references, and added several new ones.

Reviewer 3 Report

Title : Self-Adjusting Variable Neighborhood Search Algorithm for Near-Optimal k-Means Clustering
Authors : Lev Kazakovtsev, Ivan Rozhnov, Alexey Popov
-------------------------------------------------------------------------------

In this paper, the authors present a systematic approach to the construction of search algorithms in neighborhoods, formed by the use of greedy agglomerative procedures and propose a new Variable Neighborhood Search algorithm using greedy agglomerative procedures for the k-means problem. The obtained results are better than those reported in the up to date literature.

Originality / Novelty :
. This question is not original, but well defined and the way proposed to tackle this problem is original.

Significance :
. The scientific content of this paper is correct for me and deserves to be published.
. The hypotheses are correctly identified as such.
. The presented results are significant, and appropriately presented.
. The technical quality of this paper is correct for me.
. The conclusion is correctly justified and supported by the results.
. Some limits of the results obtained in this paper are mentioned. This point could be more developed. This would further improve the quality of this paper, of which I took interest and pleasure to read.

Quality of presentation :
. The abstract is clear and presents correctly the subject addressed in this paper.
. This paper contains the basic sections of a scientific paper. The correct number has to be given to the conclusion.
. No subheadings are used.
. This paper is clear, easy to follow and to read, and logically written.
. The data and analyses are appropriately presented.
. The conclusion is argumented and clear enough.

Scientific soundness :
. The subject addressed in this paper is relevant.
. The study has been correctly designed, and is technically sound.
. In my opinion, the analyses of the results are convincing.
. The data presented in this study could be completed, in order to ensure robustness to the drawn conclusions.
. The methods and software could be more deeply described, with enough details to allow another researcher to reproduce the results.

Interest to the readers :
. In my opinion, methods and conclusions of this paper seem to be interesting for the readership of the journal.

Overall evaluation :
. I think there is an overall benefit to publish this work.
. This work provides an advance towards the current knowledge, clearly highlighted in the abstract.
. The authors have addressed an already studied question, with smart experiments as well as a correct and exhaustive bibliography.
. The English language quality and style of this paper are appropriate and understandable.

As a conclusion, my suggestion to the editor is to accept this paper for publication in Computation.

References :
--------------
. 102 research references, out of which 3 self-references, giving a self-reference ratio equal to 2.94%. This is a good ratio, with no non-research reference.
. Avoid citing groups of references: [14-16], [21-23], [26-29], [39-41], [43-46], [51-53], [56-59], [70-72], [73-79], [80-82], [83-85], [51-55], [89-95]. Or if you do it anyway, please comment more any of these references.
. The chosen references are overall relevant, and are cited in the text adequately and appropriately. Nevertheless, some of them could benefit from better analysis in the literature review.
. The authors have preferred older references to more recent ones: 72 of them are more than 10 years old, and 30 of them are less than 10 years old.
. Please avoid the formulation 'et al.' in the references section: [100]. The complete list of authors deserve to be cited in this section.

Typos / Comments / Remarks :
-----------------------------------
. Line 32: can be divided can be divided --> can be divided
. Line 96: location problems, however, in large-scale --> location problems. However, in large-scale
. Lines 204-205: Repetition of the sentence.
. Line 263: |S|=r --> |S'|=r ?
. Line 264: S1 is not defined before
. Line 319: It many efficient algorithms, lmax=2. Please rewrite this sentence.
. Line 516: 4. Discussion --> 5. Discussion

Author Response

Dear Reviewer,

Thank you very much for your review.

Please find attached our detailed answers.

Reviewer 4 Report

The manuscript proposes a variable Neighborhood Search in randomized neighborhoods. Although the paper has appropriate length and informative content, several parts must be improved and written in better grammar and syntax. It would be essential if authors would consider revising the organization and composition of the manuscript, in terms of the definition/justification of the objectives, description of the method, the accomplishment of the objective, and results. The paper is generally difficult to follow. Paragraphs and sentences are not well connected. And most importantly, equations are not numbered correctly.

Furthermore, I advise considering using standard keywords to better present the research. 

Please revise the abstract according to the journal guideline. It must be under 200 words. The research question, method, and the results must be briefly communicated. The abstract must be more informative.

I suggest having four paragraphs in the introduction for; describing the concept, research gap, contribution, and the organization of the paper. The motivation has the potential to be more elaborated. You may add materials on why doing this research is essential, and what this article would add to the current knowledge, etc. The originality of the paper is not discussed well. The research question must be clearly given in the introduction, in addition to some words on the testable hypothesis. Please elaborate on the importance of this work. Please discuss if the paper suitable for broad international interest and applications or better suited for the local application? Elaborate and discuss this in the introduction.

State of the art needs improvement. A detailed description of the cited references is essential. Several recently published papers are not included in the review section. In fact, the acknowledgment of the past related work by others, in the reference list, is not sufficient. Consequently, the contribution of the paper is not clear. Furthermore, consider elaborating on the suitability of the paper and relevance to the journal. Kindly note that references cited must be up to date.    

Elaborate on the method used and why used this method.

Elaborate on the data and the results.

Elaborate on the algorithm validation, testing, and generalization.

Limitations and validation are not discussed adequately. The research question and hypothesis must be answered and discussed clearly in the discussion and conclusions. Please communicate future research. The lessons learned must be further elaborated in the conclusion by discussing the results to the community and the future impacts. What is your perspective on future research?   

There exist irrelevant references. The major improvement in English is essential.

Author Response

Dear Reviewer,

Thank you very much for your attention and comments.

Please find attached a file with our answers.

We hope that the new version of our paper is better.

Round 2

Reviewer 4 Report

Dear authors, The introduction is still not informative. The manuscript structure is not standard. The essential sections must be: introduction, materials and methods, results, discussions, and conclusions. Please reconsider all the former comments one more time. Several tables must be either removed or moved to the appendix. Please insert the acronyms table. The validation and methods description are still weak.

Author Response

Dear Reviewer,

Thank you for your efforts to make this article better.

Please find attached our answers.
